# Risk factors for third-generation cephalosporin-resistant and extended-spectrum β-lactamase-producing *Escherichia coli* carriage in domestic animals of semirural parishes east of Quito, Ecuador

**Siena L. Mitman**[1,2], **Heather K. Amato**[2], **Carlos Saraiva-Garcia**[1], **Fernanda Loayza**[1], **Liseth Salinas**[1], **Kathleen Kurowski**[3], **Rachel Marusinec**[3], **Diana Paredes**[1], **Paúl Cárdenas**[1], **Gabriel Trueba**[1], **Jay P. Graham**[2]*

**1** Instituto de Microbiología, Colegio de Ciencias Biológicas y Ambientales, Universidad San Francisco de Quito, Quito, Ecuador, **2** Division of Environmental Sciences, University of California, Berkeley School of Public Health, Berkeley, California, United States of America, **3** Division of Infectious Diseases and Vaccinology, University of California, Berkeley School of Public Health, Berkeley, California, United States of America

* jay.graham@berkeley.edu

**Data Availability Statement:** The dataset used in this study is available here: Mitman, Siena (2021),

## Abstract

Extended-spectrum β-lactamase (ESBL)-producing and other antimicrobial resistant (AR) *Escherichia coli* threaten human and animal health worldwide. This study examined risk factors for domestic animal colonization with ceftriaxone-resistant (CR) and ESBL-producing *E. coli* in semirural parishes east of Quito, Ecuador, where small-scale food animal production is common. Survey data regarding household characteristics, animal care, and antimicrobial use were collected from 304 households over three sampling cycles, and 1195 environmental animal fecal samples were assessed for *E. coli* presence and antimicrobial susceptibility. Multivariable regression analyses were used to assess potential risk factors for CR and ESBL-producing *E. coli* carriage. Overall, CR and ESBL-producing *E. coli* were detected in 56% and 10% of all fecal samples, respectively. The odds of CR *E. coli* carriage were greater among dogs at households that lived within a 5 km radius of more than 5 commercial food animal facilities (OR 1.72, 95% CI 1.15–2.58) and lower among dogs living at households that used antimicrobials for their animal(s) based on veterinary/pharmacy recommendation (OR 0.18, 95% CI 0.04–0.96). Increased odds of canine ESBL-producing *E. coli* carriage were associated with recent antimicrobial use in any household animal (OR 2.69, 95% CI 1.02–7.10) and purchase of antimicrobials from pet food stores (OR 6.83, 95% CI 1.32–35.35). Food animals at households that owned more than 3 species (OR 0.64, 95% CI 0.42–0.97), that used antimicrobials for growth promotion (OR 0.41, 95% CI 0.19–0.89), and that obtained antimicrobials from pet food stores (OR 0.47, 95% CI 0.25–0.89) had decreased odds of CR *E. coli* carriage, while food animals at households with more than 5 people (OR 2.22, 95% CI 1.23–3.99) and located within 1 km of a commercial food animal facility (OR 2.57, 95% CI 1.08–6.12) had increased odds of ESBL-producing *E. coli*

MITMAN_PRISA_AMRinAnimalsEcuador, Dryad, Dataset, https://doi.org/10.6078/D1ZM6F.

**Funding:** This research was funded by the National Institutes of Health under Award Number R01AI135118 (all authors) and the Fogarty International Center Global Heath Equity Scholars Program, NIH FIC D43 TW010540 (PC, SM). The funders had no role in study design, data collection and analysis, decision to publish, or preparation of the manuscript.

**Competing interests:** The authors have declared that no competing interests exist.

carriage. Together, these results highlight the complexity of antimicrobial resistance among domestic animals in this setting.

## Introduction

Historically, third-generation cephalosporin antimicrobials have been used clinically to treat infections caused by Gram-negative bacteria in both human and veterinary medicine [1]. However, third-generation cephalosporin resistant (3CGR) bacteria, including those that can produce extended-spectrum β-lactamases (ESBLs), are becoming increasingly common [2–5]. ESBL-producing Enterobacteriaceae now represent a significant threat to both human and animal health worldwide [1, 6–8]. *Escherichia coli* is an important ESBL-producing species because of its ability to shift its antimicrobial resistance (AMR) phenotypes in environments outside the host, colonize a wide variety of species, and evolve from a commensal, antimicrobial-susceptible bacterium into a resistant, pathogenic organism [2, 9–12]. Commensal *E. coli*, often used as an indicator of the selective antimicrobial pressures on Enterobacteriaceae, is common among the intestinal tracts of warm-blooded hosts and can serve as a reservoir for ESBL genes, actively participating in horizontal gene transfer to other bacteria, including pathogenic ones [2, 13–17].

Given the widespread prevalence of ESBL-producing and other antimicrobial resistant (AR) bacteria among humans, the environment, and diverse animal species, there is now a growing consensus that addressing this concern requires a One Health approach, in which the interconnected roles of humans, animals, and the environment are considered [15, 18–21]. Surveillance and control is particularly pressing in the agricultural sector, where antimicrobials are often used prophylactically to prevent disease and promote growth in food animal production [9, 22, 23]. There is widespread documentation of ESBL-producing *E. coli* in food animals around the globe [24–32]. Transmission of ESBL-producing *E. coli* from food animals to humans may occur through direct contact or human consumption of meat and animal byproducts [33]. Environmental reservoirs, such as waterways or soil, contaminated with domestic animal waste have also been implicated in the spread of ESBL-producing *E. coli*, ESBL resistance genes, and mobile genetic elements that facilitate horizontal gene transfer [13, 33–37]. Companion animals such as cats and dogs can also be colonized with ESBL-producing *E. coli*, and may contribute to transmission to humans through similar mechanisms [38–43]. In some settings, companion animals could also act as intermediate ESBL-producing *E. coli* hosts between food animals and humans [25, 28].

Addressing the threat of 3GCR, ESBL-producing and other AR *E. coli* in domestic animals requires an understanding of their risk factors in specific contexts. Most studies assessing such risks have focused on commercial food animal production settings, where increased risk in varying food animals has been linked to a wide variety of practices, including antimicrobial use and sanitation practices [25, 27, 28, 44–46]. Studies assessing risk factors for companion animal colonization with ESBL-producing and other AR *E. coli* have focused primarily on dogs and identified risk factors such as consumption of raw meat or poultry, previous hospitalization, treatment with antimicrobials, and contact with livestock [39, 43, 47–49]. Unlike *E. coli* with other AMR phenotypes, ESBL-producing *E. coli* has been consistently associated with animal exposure to antimicrobials [46, 50, 51].

The role of small-scale food animal production in the transmission of ESBL-producing and other AR *E. coli*, however, is largely unexplored, despite the widespread prevalence of small-

scale or backyard food animals worldwide. Such practices are particularly common in low- and middle-income countries (LMICs) where they can play an important role in food and nutrition security, women's empowerment, and nutrient recycling and utilization [52–55]. In many LMICs, antimicrobials are often available over the counter and used with limited veterinary oversight to promote animal growth and prevent disease [23, 56–59]. There is thus increasing concern that food animals from small-scale production settings may contribute to the prevalence and transmission of ESBL-producing and other AR bacteria, though to what extent is unknown [23]. The research that exists regarding risk factors for AR *E. coli* carriage in animals in small-scale production settings suggests that specific risk factors contributing to domestic animal colonization with ESBL-producing and other AR *E. coli* are likely context-dependent, warranting closer attention in specific settings where small-scale food animal production occurs [26, 29, 35].

In Ecuador, small-scale food animal caretakers commonly use over-the-counter antimicrobials for their animals [56, 57, 60]. In communities outside of Quito, horizontal transfer of AMR genes and mobile genetic elements is thought to play a dominant role in ESBL-producing *E. coli* transmission between domestic animals and humans [13, 61]. While risk factors for colonization with ESBL-producing *E. coli* have been identified for children in the region, specific household characteristics and antibiotic knowledge, attitudes, and practices (KAP) contributing to domestic animal colonization with 3GCR and ESBL-producing *E. coli* have yet to be explored [62]. Identifying such risk factors could help direct future mitigation and prevention strategies. The current study thus aimed to 1) estimate the prevalence of resistance to ceftriaxone (a third-generation cephalosporin) and ESBL-producing *E. coli* among domestic animal fecal samples in semirural parishes east of Quito, Ecuador and 2) assess the household characteristics, animal care practices, and antibiotic KAP that contribute to domestic animal carriage of ceftriaxone-resistant and ESBL-producing *E. coli* in the region.

## Materials and methods

### Study location and recruitment

This analysis was conducted as part of a larger repeated measures study in 7 semirural parishes east of Quito, Ecuador. There are 32 urban parishes and 33 rural or suburban parishes that make up Quito. The 7 parishes selected for this study were considered to be representative of communities that live near large major metropolitan areas while maintaining many rural practices, including small-scale food animal production. Parishes were also selected based on their proximity to the Universidad San Francisco de Quito, as samples needed to be analyzed on the same day as they were collected.

Households from these 7 parishes were randomly enrolled in the parent study with the following inclusion criteria: 1) there was a primary household caregiver at least 18 years of age present; 2) there was a child living at the household between 6 months and 5 years of age; 3) written informed consent was provided by the primary childcare provider. Households reporting any animal ownership and with at least one animal fecal sample collected were selected for inclusion in this study's analysis. Data collection occurred in three separate cycles of 20 weeks in duration from July 2018 to May 2019. New samples were collected from each household enrolled in the study in the first cycle in each subsequent cycle. When households were lost to follow-up, new households were enrolled in the study based on the same inclusion criteria.

### Knowledge, attitudes, and practices (KAP) survey

Trained study personnel administered a previously validated antibiotic KAP survey to caretakers at each household one time per cycle prior to fecal sample collection. The survey included

questions regarding animal care, sanitation, and feeding practices as well as proximity to commercial food animal (livestock and poultry) facilities, antimicrobial use and knowledge, and socioeconomic factors such as education and asset ownership, which were used as indicators of wealth. Survey questions were derived from similar KAP-based studies [63, 64] and discussed and modified with local stakeholders to ensure that the questions would be appropriate for the study's specifications. Validation was accomplished by first establishing face-validity through review by field staff living in the communities involved. Field staff evaluated survey questions and determined whether they successfully captured the intended purpose. The survey was then pilot tested with community residents to address issues of comprehension, and revisions were made. The interview team then field-tested questions to determine whether they were culturally appropriate and relevant, and surveys were adjusted based on feedback during the first four weeks of sampling.

Surveys were written in English, translated to Spanish, and translated back to English to assure accurate translation. Surveys were conducted in Spanish using tablets and Open Data Kit (ODK) Collect software, version 1.22.3 (getodk.org). Survey data are stored on a secure server and were downloaded for processing and de-identification before beginning this analysis. Survey templates used are available in Spanish and English in the S1 and S2 Surveys.

## Fecal sample collection

Fresh animal fecal samples were aseptically collected from each household's environment. Study personnel collected environmental animal fecal samples from all species present at the household, when possible, in separate sterile collection tubes. When more than one sample per animal species was collected from a household at the same time point, feces from the same species were pooled into one collection tube. Samples were labeled and transported back to the laboratory at the Universidad San Francisco de Quito in a chilled container at approximately 4˚C for processing within 5 hours of collection.

## *E. coli* identification

*E. coli* was isolated as described previously [13]. Briefly, fecal samples were incubated at 37 ˚C overnight on a selective media with MacConkey agar (Difco, Sparks, Maryland) and ceftriaxone (2 mg/L), a third-generation cephalosporin. Colonies that grew on this supplemented agar were considered Ceftriaxone-resistant (CR). If present, five lactose-positive colonies were randomly selected from each sample, cultured in Trypticase Soy Broth (Difco, Sparks, Maryland) with 15% glycerol, and preserved at -80 ˚C for further analysis.

## Antimicrobial susceptibility testing

To assess AMR phenotypes, one isolate per sample was thawed and cultured overnight at 37˚C on MacConkey agar (Difco, Sparks, Maryland) with ceftriaxone (2 mg/L). Each isolate was also cultured on Chromocult coliform agar (Merck KGaA, Darmstadt, Germany) at 37˚C for 16–20 hours for assessment of β-D-glucuronidase activity to confirm *E. coli* identification [65]. To assess antimicrobial susceptibility phenotype, Kirby-Bauer disk diffusion testing was used. Isolates were streaked on Mueller-Hinton agar plates (Difco, Sparks, Maryland) and incubated overnight at 37 ˚C for 16–20 hours with antimicrobial disks. Antimicrobials evaluated included: gentamicin (GM; 10 μg), imipenem (IMP; 10 μg), ceftazidime/clavulanic acid (CAZ-CLA; 30/10 μg), trimethoprim-sulfamethoxazole (SXT; 1.25/23.75 μg), ceftazidime (CAZ; 30 μg), cefepime (FEP; 30 μg), ciprofloxacin (CIP; 5 μg), amoxicillin/clavulanic acid (AMC; 20/10 μg), cefazolin (CZ; 30 μg), ampicillin (AM; 10 μg), cefotaxime (CTX; 30 μg), and

tetracycline (TE; 30 μg). *E. coli* 25922 was used as a control strain in addition to a negative control.

Clinical resistance was interpreted as described by the Clinical and Laboratory Standards Institute [66]. The CLSI human medicine guidelines were used given the public health focus of this study. ESBL-producing *E. coli* were defined by double disk synergy, in which the isolate was resistant to ceftazidime, and the difference in the inhibition zone diameter of ceftazidime/ clavulanic acid and ceftazidime was greater than 5 mm [66]. Isolates with intermediate resistance were considered susceptible for the sake of binary outcome analysis. Third-generation cephalosporin multidrug resistant (3GCR-MDR) *E. coli* was defined as *E. coli* resistant to ceftriaxone and at least three classes of antimicrobials, and third-generation cephalosporin extensively drug resistant (3GCR-XDR) *E. coli* was defined as *E. coli* resistant to ceftriaxone and at least five classes of antimicrobials [10]. Antimicrobial classes assessed include the following: penicillins +/- β-lactamase inhibitors (ampicillin and amoxicillin/ clavulanic acid), aminoglycosides (gentamycin), cephalosporins +/- β-lactamase inhibitors (cefazolin, ceftazidime, cefotaxime, cefepime, and ceftazidime/clavulanic acid), carbapenems (imipenem), fluoroquinolones (ciprofloxacin), tetracyclines (tetracycline), and folate pathway antagonists (trimethoprim-sulfamethoxazole).

## Statistical analyses

The primary outcomes of this analysis were whether or not a fecal sample from a dog or food animal (including cows, guinea pigs, pigs, rabbits, sheep, horses, llamas, chickens, ducks, or other poultry) was positive for: 1) CR *E. coli* and 2) ESBL-producing *E. coli*. Secondary outcomes included were whether or not a fecal sample was positive for: 1) CR and MDR *E. coli* (resistant to ceftriaxone and 3 or more classes of antimicrobials, called 3GCR-MDR) and 2) CR and XDR *E. coli* (resistant to ceftriaxone and 5 or more classes of antimicrobials, called 3GCR-XDR). When an individual household had samples collected across multiple cycles, each time point was treated as a separate outcome and compared to risk factors at the household during that same cycle. CR, ESBL-producing, 3GCR-MDR, and 3GCR-XDR *E. coli* prevalence among fecal samples was calculated as the proportion of animal fecal samples positive for each AMR phenotype out of all fecal samples collected across three cycles. CR, ESBL-producing, 3GCR-MDR, and 3GCR-XDR prevalence was also calculated across all households included in the study.

Potential risk factors analyzed in this study included household and caregiver characteristics such as demographic variables, animal ownership, animal care practices, and antibiotic knowledge, attitudes, and practices (KAP). Household wealth was determined by a principal component analysis using household asset data collected in the survey. Household assets included: car(s), television(s), cable television(s), computer(s), internet, house(s), and land. A wealth index score was created for each household in the parent study, and scores were divided by tertiles into low, medium, and high categories. A livestock unit variable was also created based on Eurostat guidelines and adjusted to include food animals relevant in this setting. The use of livestock units is an approach recommended by several agricultural organizations, including the United Nations Food and Agricultural Organization (UN FAO), to account for the size of animals and the relative fecal waste produced and to allow comparison of households with different types of animals (i.e., 1 cow versus 100 guinea pigs). The number of livestock units (LU) was calculated using the following equation [67]:

$$LU = (0.01)(N_{chickens}) + (0.30)(N_{pigs}) + (0.80)(N_{cattle}) + (0.10)(N_{sheep}) + (0.10)(N_{goats})$$
$$+ (0.02)(N_{rabbits}) + (0.01)(N_{guinea\ pigs}) + (0.03)(N_{ducks}) + (0.03)(N_{quail})$$

where N was the number of animals of the given species owned by the household. For all risk factors, continuous variables were categorized using cut points selected based on sample distributions, with consideration for future policy or intervention relevance. A Chi-square goodness of fit test was used for each variable to identify significant differences across study cycles, with significance set at a level of $\alpha$ = 0.05.

Univariable and multivariable logistic regression models were created to estimate the associations between each risk factor and outcome variable. Multivariable regressions included pre-specified confounding variables for each risk factor-outcome relationship, determined using directed acyclic graphs and existing literature (S1 Table). Co-linear variables were not included as confounders. Odds ratios (OR) with robust 95% confidence intervals (CI) were calculated for each risk factor and outcome of interest. Statistical significance was assessed at $\alpha$ = 0.05. Robust standard errors were estimated using generalized estimating equations (GEE) and an exchangeable working correlation to adjust for unbalanced data and repeated measures at the household level. All analyses were conducted with R Studio version 1.3 (R Core Team, 2020) using the following packages: *tidyverse* [68], *dplyr* [69], *knitr* [70], *ggplot2* [71], *gtools* [72], *psych* [73], *table1* [74], *broom* [75], *geepack* [76].

## Ethics

The study protocol was approved by the University of California, Berkeley Committee for Protection of Human Subjects (IRB# 2019-02-11803), the Bioethics Ethical Committee at the Universidad San Francisco de Quito (#2017-178M), and the Ecuadorian Health Ministry (#MSPCURI000243-3).

## Results

### Study population demographics and animal ownership

We collected 1195 domestic animal fecal samples from 304 households across the three 20-week long cycles included in this study. Among the 212 households that participated in cycle one, 67 were lost to follow-up and replaced by 59 new households in cycle two. Of the 204 households that participated in cycle two, 60 were lost to follow-up and replaced by 47 households in cycle three, including 14 households that had previously participated in cycle one. Among all participating households, 107 participated in all three cycles.

A summary of household characteristics by study cycle is shown in Table 1. The majority of survey respondents were female (90%, 93%, 94% of households in cycles one, two, and three, respectively), and the median ages of respondents for cycles one, two, and three were 26, 28, and 30 years, respectively. Most households (64%, 67%, 65%) were medium-sized, with 4–6 members, and the majority of respondents (71%, 69%, 66%) had obtained a high school degree or higher. The most common household wealth category for cycles one and three was medium wealth (49% of households in cycle one, 40% of households in cycle three), while that of cycle two was low wealth (47% of households in cycle two). Households varied in distance to the nearest commercial food animal facility, which were primarily poultry (97% of households in each cycle) and rarely swine (2%, 3%, 3%), and in density of commercial food animal facilities within a 5 km radius. Though most households (81%, 78%, 77%) were located 2 km or closer to the nearest food animal facility, the majority of study respondents (69%, 69%, 73%) did not report smelling poultry odors at their homes.

Dogs were owned by almost all households in each cycle (98%, 93%, 94% of households in cycles one, two, and three, respectively), and chickens were owned by almost half of all households (42%, 43%, 45%). Approximately one half of study households owned 1–5 animals (56%, 53%, 54%), while the majority of households owned 3 or fewer total species (79%, 76%, 73%)

**Table 1. Household and caregiver characteristics by data collection cycle.**

| Characteristic | Cycle 1 (212 households) | Cycle 2 (204 households) | Cycle 3 (191 households) | p-value[1] |
|---|---|---|---|---|
| *Caregiver age (n = 212, 204, 191)* | | | | |
| <30 years old | 127 (60%) | 112 (55%) | 93 (49%) | 0.08 |
| ≥30 years old | 85 (40%) | 92 (45%) | 98 (51%) | |
| *Respondent sex (n = 211, 204, 191)* | | | | |
| Female | 190 (90%) | 189 (93%) | 179 (94%) | 0.39 |
| Male | 21 (10%) | 14 (7%) | 12 (6%) | |
| Transgender | 0 | 1 (0.4%) | 0 | |
| *Household wealth (n = 212, 204, 191)* | | | | |
| Low | 43 (20%) | 96 (47%) | 48 (25%) | 0.00 |
| Medium | 103 (49%) | 44 (22%) | 76 (40%) | |
| High | 66 (31%) | 64 (31%) | 67 (35%) | |
| *Household size (n = 212, 204, 191)* | | | | |
| Small (1–3) | 55 (26%) | 46 (23%) | 42 (22%) | 0.84 |
| Medium (4–6) | 135 (64%) | 136 (67%) | 125 (65%) | |
| Large (7–11) | 22 (10%) | 22 (11%) | 24 (13%) | |
| *Highest level of caregiver education (n = 212, 204, 191)* | | | | |
| Elementary School | 60 (28%) | 63 (31%) | 65 (34%) | 0.62 |
| High School | 130 (61%) | 119 (58%) | 102 (53%) | |
| University | 22 (10%) | 22 (11%) | 24 (13%) | |
| *Nearest food animal facility type (n = 211, 204, 191)* | | | | |
| Poultry | 206 (97%) | 198 (97%) | 186 (97%) | 0.94 |
| Swine | 5 (2%) | 6 (3%) | 5 (3%) | |
| *Proximity to nearest commercial food animal facility (n = 212, 204, 191)* | | | | |
| <1 km | 91 (43%) | 83 (41%) | 78 (41%) | 0.84 |
| 1–2 km | 81 (38%) | 76 (37%) | 68 (36%) | |
| >2 km | 40 (19%) | 45 (22%) | 45 (24%) | |
| *Commercial food animal facilities within 5 km (n = 211, 204, 191)* | | | | |
| 0–5 | 53 (25%) | 51 (25%) | 47 (25%) | 0.91 |
| 6–10 | 50 (24%) | 45 (22%) | 40 (21%) | |
| 10–20 | 61 (29%) | 66 (32%) | 55 (29%) | |
| >20 | 47 (22%) | 42 (21%) | 49 (26%) | |
| *Commercial poultry odors detected by respondent (n = 211, 204, 191)* | | | | |
| No/don't know | 147 (69%) | 140 (69%) | 140 (73%) | 0.55 |
| Yes | 65 (31%) | 64 (31%) | 51 (27%) | |
| *Number of species at household (n = 209, 204, 191)* | | | | |
| 1 | 91 (44%) | 94 (46%) | 83 (43%) | 0.58 |
| 2–3 | 74 (35%) | 61 (30%) | 57 (30%) | |
| 4–9 | 44 (21%) | 49 (24%) | 51 (27%) | |
| *Number of animals at household (n = 209, 204, 191)* | | | | |
| 1–5 | 119 (56%) | 109 (53%) | 104 (54%) | 0.33 |
| 6–20 | 46 (22%) | 62 (30%) | 48 (25%) | |
| >20 | 44 (21%) | 33 (16%) | 39 (20%) | |
| *Number of food animals at household (n = 210, 204, 191)* | | | | |
| None | 107 (50%) | 96 (47%) | 95 (50%) | 0.78 |
| 1–10 | 49 (23%) | 57 (28%) | 44 (23%) | |
| >10 | 54 (25%) | 51 (25%) | 52 (27%) | |
| *Livestock units[2] at household (n = 210, 204, 191)* | | | | |

*(Continued)*

**Table 1.** (Continued)

| Characteristic | Cycle 1 (212 households) | Cycle 2 (204 households) | Cycle 3 (191 households) | p-value[1] |
|---|---|---|---|---|
| None | 107 (50%) | 96 (47%) | 95 (50%) | 0.73 |
| >0 and ≤ 1 | 77 (36%) | 86 (42%) | 70 (37%) | |
| >1 | 26 (12%) | 22 (11%) | 26 (14%) | |
| *Species of animals owned at household* | | | | |
| Dogs *(n = 212, 204, 191)* | 208 (98%) | 189 (93%) | 179 (94%) | 0.03 |
| Cats *(n = 211, 204, 191)* | 64 (30%) | 50 (25%) | 56 (29%) | 0.37 |
| Chickens *(n = 212, 204, 191)* | 90 (42%) | 87 (43%) | 85 (45%) | 0.90 |
| Guinea pigs *(n = 211, 204, 191)* | 36 (17%) | 54 (26%) | 42 (22%) | 0.07 |
| Pigs *(n = 211, 204, 191)* | 27 (13%) | 35 (17%) | 32 (17%) | 0.40 |
| Rabbits *(n = 212, 204, 191)* | 20 (9%) | 25 (12%) | 31 (16%) | 0.12 |
| Ducks *(n = 212, 204, 191)* | 18 (8%) | 21 (10%) | 21 (11%) | 0.68 |
| Cows *(n = 212, 204, 191)* | 15 (7%) | 21 (10%) | 19 (10%) | 0.46 |
| Other[3] *(n = 212, 204, 191)* | 18 (8%) | 11 (5%) | 16 (8%) | 0.40 |

[1]Chi-square goodness of fit test used to determine p-value with significance at α = 0.05.

[2]Livestock units = (0.01) (number of chickens) + (0.30) (number of pigs) + (0.80) (number of cattle) + (0.10) (number of sheep) + (0.10) (number of goats) + (0.02) (number of rabbits) + (0.01) (number of guinea pigs) + (0.03) (number of ducks) + (0.03) (number of quail).

[3]Other = Quail, sheep, goats, and others (non-specified).

and 1 or fewer livestock units (86%, 89%, 87%). Of the households that owned chickens, approximately 66% owned 10 or fewer birds. Among households that owned food animals, 52% owned fewer than 10 animals, confirming this setting as one of primarily small-scale or backyard food animal ownership.

## Animal care and antibiotic knowledge, attitudes, and practices (KAP)

A summary of household animal care and antibiotic KAP by study cycle is shown in Table 2. Most households in this study (83%, 85%, 91% of households in cycles one, two, and three, respectively) did not report giving antibiotics to any of their animals in the past 6 months. Most households (80%, 83%, 84%) also did not report using other medications or vitamins in their animals in the past 6 months. Access to veterinary care, defined by a household's self-reported answer to whether or not they had veterinary access, was limited to approximately 10% or fewer households in each cycle (9%, 10%, 7%). When asked about frequency of antibiotic use for animals, households that used antibiotics most often reported giving them as needed (83%, 87%, 83% of households that used antibiotics in cycles one, two, and three respectively). When asked about their motivations for giving antibiotics, households most often reported using them for growth promotion (reported by 36%, 23%, 28% of households that used antibiotics) and illness prevention (reported by 22%, 17%, 44% of households that used antibiotics). All households that reported having veterinary access reported obtaining antibiotics from a veterinarian, while those that did not report veterinary access most often obtained antibiotics from a pet food store (reported by 32%, 24%, 18% of households that used antibiotics).

Approximately one third of households in each cycle (38%, 35%, 31% of households in cycles one, two, and three, respectively) reported use of commercial animal feed while animal consumption of river or irrigation water in the past 3 weeks was limited (reported by 9%, 8%, and 9% of households in cycles one, two, and three, respectively). Households most often

**Table 2. Household animal care and antibiotic knowledge, attitudes, and practices (KAP) by data collection cycle.**

| Animal care and antibiotic KAP | Cycle 1 (212 households) | Cycle 2 (204 households) | Cycle 3 (191 households) | p-value[1] |
|---|---|---|---|---|
| *Animal(s) given antibiotics in past 6 months (n = 203, 201, 190)* | 36 (17%) | 30 (15%) | 18 (9%) | 0.06 |
| *Other medications/vitamins given in past 6 months (n = 210,203,191)* | 42 (20%) | 35 (17%) | 31 (16%) | 0.59 |
| *Frequency of antibiotic use (n = 30, 30, 18)[2]* | | | | |
| As needed | 25 (83%) | 26 (87%) | 15 (83%) | 0.46 |
| Routinely | 3 (10%) | 4 (13%) | 3 (17%) | |
| Don't know | 2 (7%) | 0 (0%) | 0 (0%) | |
| *Reasons for antibiotic use[2]* | | | | |
| Growth promotion (n = 32, 29, 18) | 13 (36%) | 7 (23%) | 5 (28%) | 0.30 |
| Illness prevention (n = 32, 30, 18) | 8 (22%) | 5 (17%) | 8 (44%) | 0.10 |
| Illness treatment (n = 32, 30, 16) | 5 (14%) | 2 (7%) | 4 (22%) | 0.22 |
| Pharmacy/Vet recommended (n = 32, 30, 18) | 3 (8%) | 5 (17%) | 5 (28%) | 0.24 |
| *Antibiotic source (n = 34, 29, 17)[2]* | | | | |
| Veterinarian | 20 (59%) | 21 (72%) | 14 (82%) | 0.41 |
| Pet food store | 11 (32%) | 7 (24%) | 3 (18%) | |
| Other/don't know | 3 (9%) | 1 (3%) | 0 (0%) | |
| *Access to veterinary care (n = 201,200, 190)* | 20 (9%) | 21 (10%) | 14 (7%) | 0.53 |
| *Animal(s) consumed river or irrigation water in past 3 weeks (n = 212, 204, 191)* | 19 (9%) | 16 (8%) | 18 (9%) | 0.85 |
| *Animal(s) fed commercial feed (n = 202, 203, 189)* | 80 (38%) | 72 (35%) | 60 (31%) | 0.27 |
| *Household member worked with animals or in animal/animal by-product processing outside the home in past 6 months (n = 212, 204, 191)* | 71 (33%) | 52 (25%) | 32 (17%) | 0.001 |
| *Household member worked with human or animal feces outside the home in past 6 months (n = 212, 204, 191)* | 22 (10%) | 15 (7%) | 28 (15%) | 0.06 |
| *Household member took antibiotics in past 3 months (n = 72, 68, 52)[3]* | 65 (90%) | 57 (84%) | 35 (67%) | 0.01 |
| *Animals allowed inside the home (n = 210, 204, 190)* | 95 (45%) | 73 (36%) | 104 (54%) | 0.001 |
| *Animals allowed near children (n = 210, 204, 191)* | 113 (53%) | 113 (55%) | 131 (69%) | 0.005 |
| *Animal feces management (n = 197, 195, 186)* | | | | |
| Place in trash | 96 (49%) | 101 (52%) | 85 (46%) | 0.009 |
| Leave in yard to decompose | 29 (15%) | 39 (20%) | 49 (26%) | |
| Store and place on land/ Use for crops as fertilizer | 62 (31%) | 48 (25%) | 36 (19%) | |
| Other/don't know | 10 (5%) | 7 (4%) | 16 (9%) | |
| *Answer to whether or not antibiotics kill bacteria (n = 212, 202, 191)* | | | | |
| "Yes"/Correct | 78 (37%) | 83 (41%) | 66 (35%) | 0.33 |
| "No"/Incorrect | 45 (21%) | 32 (16%) | 46 (24%) | |
| Don't know | 89 (42%) | 87 (43%) | 79 (41%) | |
| *Answer to whether or not antibiotics kill viruses (n = 211, 203, 191)* | | | | |
| "No"/Correct | 44 (21%) | 30 (15%) | 51 (27%) | |
| "Yes"/Incorrect | 83 (39%) | 84 (41%) | 58 (30%) | 0.03 |
| Don't know | 84 (40%) | 89 (44%) | 82 (43%) | |

[1]Chi-square goodness of fit test used to determine p-value with significance at $\alpha = 0.05$.

[2]Questions regarding motivation for antibiotic use and antibiotic source were only answered by those caregivers that reported using antibiotics for their animal(s).

[3]Household member use of antibiotics was determined based on caregiver response to whether or not their child in the study had taken antibiotics in the past 3 months and whether or not a household member had taken antibiotics in the past 3 months, the latter of which was only asked to those who reported having a household member with an illness or infection in the past 3 months.

discarded animal fecal waste in the trash (reported by 49%, 52%, and 46% of households in cycles one, two, and three, respectively), though households also reported leaving feces in the yard to decompose (15%, 20%, 26%), storing feces then using it as fertilizer or otherwise placing it on their land (31%, 25%, 19%), or utilizing other methods of disposal (5%, 4%, 9%). Most households (90%, 93%, 85%) did not have a member working outside the home with human or animal feces, though 33%, 25%, and 17% of households across the three sampling cycles reported having a member that had worked with animals or in animal or animal-byproduct processing in the past 6 months. Most households (69%, 72%, 82%) did not report having a human member that had taken antibiotics in the past 3 months. When asked if antibiotics kill bacteria, 37%, 41%, and 35% of survey respondents correctly responded "yes" across the three cycles. When asked if antibiotics kill viruses, 21%, 15%, and 27% of respondents in cycles one, two, and three, respectively, correctly answered "no."

## Sample characteristics and *E. coli* antimicrobial resistance (AMR) phenotypes

Table 3 describes the fecal sample characteristics and AMR phenotypes among all samples analyzed by species. Of the 1195 domestic animal fecal samples collected in this study, the majority came from dogs (n = 555 fecal samples) and chickens (n = 244). Additional fecal samples were collected from guinea pigs (n = 110), pigs (n = 76), rabbits (n = 68), cows (n = 48), ducks (n = 44), other poultry, including geese (n = 11), pigeons (n = 10), and quail (n = 7), llamas (n = 2), cats (n = 1), and hamsters (n = 1). Llamas, cats, and hamsters were referred to as "other" for prevalence analyses.

CR *E. coli* were isolated from 56% (670/1195) and ESBL-producing *E. coli* from 10% (123/1195) of all animal fecal samples. Both outcomes were most common in pigs (CR 72%; ESBL

**Table 3. CR, ESBL-producing, 3GCR-MDR, and 3GCR-XDR *E. coli* isolated from domestic animal fecal samples by species.**

| Species | Fecal samples | Households sampled | CR isolates (%)[1] | ESBL isolates (%)[1] | 3GCR-MDR isolates (%)[1] | 3GCR-XDR isolates (%)[1] | Average total classes of resistance[2] |
|---|---|---|---|---|---|---|---|
| Dogs | 555 | 288 | 376 (68%) | 71 (13%) | 348 (63%) | 183 (33%) | 2.9 ± 2.2 |
| Chickens | 244 | 148 | 170 (70%) | 29 (12%) | 151 (62%) | 65 (27%) | 2.8 ± 2.1 |
| Guinea pigs | 110 | 64 | 1 (1%) | 0 (0%) | 1 (1%) | 0 (0%) | 0.04 ± 0.38 |
| Pigs | 76 | 47 | 55 (72%) | 12 (16%) | 50 (66%) | 21 (28%) | 3.0 ± 2.20 |
| Rabbits | 68 | 44 | 6 (9%) | 2 (3%) | 4 (6%) | 0 (0%) | 0.28 ± 0.94 |
| Ducks | 44 | 31 | 36 (82%) | 4 (9%) | 34 (77%) | 10 (23%) | 3.2 ± 1.80 |
| Cows | 48 | 29 | 17 (35%) | 1 (2%) | 13 (27%) | 3 (6%) | 1.3 ± 1.80 |
| Other poultry[3] | 28 | 20 | 6 (21%) | 3 (11%) | 6 (21%) | 2 (7%) | 0.82 ± 1.70 |
| Sheep | 10 | 6 | 1 (10%) | 0 (0%) | 1 (10%) | 0 (0%) | 0.40 ± 1.30 |
| Horse | 7 | 7 | 1 (14%) | 1 (14%) | 0 (0%) | 0 (0%) | 0.29 ± 0.76 |
| Other[4] | 4 | 4 | 0 (0%) | 0 (0%) | 0 (0%) | 0 (0%) | 0.00 ± 0.00 |
| Total | 1195 | 304 | 670 (56%) | 123 (10%) | 608 (51%) | 284 (24%) | 2.3 ± 2.3 |

[1]Percentage of species-specific fecal samples.

[2]Antimicrobial classes include penicillins +/- β-lactamase inhibitors (ampicillin and amoxicillin/ clavulanic acid), aminoglycosides (gentamycin), cephalosporins +/- β-lactamase inhibitors (cefazolin, ceftazidime, cefotaxime, cefepime, and ceftazidime/clavulanic acid), carbapenems (imipenem), fluoroquinolones (ciprofloxacin), tetracyclines (tetracycline), and folate pathway antagonists (trimethoprim-sulfamethoxazole).

[3]Other poultry = geese (11), pigeon (10), and quail (7).

[4]Other = llamas (2), cat (1), and hamster (1).

16%), dogs (CR 68%; ESBL 13%), chickens (CR 70%; ESBL 12%), and ducks (CR 82%; ESBL 9%). Overall, 87% (264/304) of individual households had at least one CR-positive sample identified throughout the course of this study (i.e. household had at least one CR-positive fecal sample collected in at least one sampling cycle), including CR *E. coli* confirmed from a dog at 82% of households that owned dogs (241/295) and from a food animal at 73% (130/177) of households that owned food animals. ESBL-positive samples were identified at least once at 32% (96/304) of households, including ESBL-producing *E. coli* confirmed from a dog at 22% of households that owned dogs (64/295) and from a food animal at 25% (44/177) of households that owned food animals.

On average, *E. coli* from animal fecal samples were resistant to 2.3 classes of antimicrobials. Animal species with *E. coli* resistant to the highest average number of classes included ducks (3.2 ± 1.80), pigs (3.0 ± 2.20), dogs (2.9 ± 2.2), and chickens (2.8 ± 2.1). 3GCR-MDR and 3GCR-XDR *E. coli* were isolated from 51% (609/1195) and 24% (284/1195) of all fecal samples and identified in animal fecal samples at least once at 82% (250/304) and 59% (178/304) of households, respectively. 3GCR-MDR and 3GCR-XDR *E. coli* were most commonly isolated from fecal samples from ducks (3GCR-MDR 77% of duck fecal samples; 3GCR-XDR 23%), pigs (3GCR-MDR 66%; 3GCR-XDR 28%), dogs (3GCR-MDR 63%; 3GCR-XDR 33%), and chickens (3GCR-MDR 62%; 3GCR-XDR 27%). Overall, 77% (227/295) and 49% (146/295) of all households that owned dogs had at least one 3GCR-MDR or 3GCR-XDR positive *E. coli* fecal sample, respectively, collected from a dog, while 69% (122/177) and 44% (77/177) of all households that owned food animals had at least one 3GCR-MDR or 3GCR-XDR positive *E. coli* fecal sample collected from a food animal.

CR *E. coli* isolates were most commonly resistant to cefazolin, a first-generation cephalosporin (99.7% of all *E. coli* isolates), ampicillin, a penicillin (99.6%), cefotaxime, a third-generation cephalosporin (96.7%), tetracycline, a tetracycline (82.7%), and trimethoprim-sulfamethoxazole, a folate pathway antagonist (65.4%). One isolate (0.15%), originating from a chicken fecal sample in cycle two, was resistant to imipenem, a carbapenem and last-line antibiotic in human medicine. Phenotypic resistance patterns by species were similar across cycles with some minor variations (S2 Table).

## Risk factors for ceftriaxone-resistant (CR) and ESBL-producing *E. coli* in dogs

Adjusted odds ratios for CR and ESBL-producing *E. coli* carriage in dogs based on household characteristic risk factors are summarized in Table 4. Adjusted odds ratios showed that households with greater than 5 commercial food animal facilities within a 5 km radius had higher odds of CR and ESBL-producing *E. coli*, though this relationship was only significant for CR *E. coli* (OR 1.72, 95% CI 1.15–2.58). Dogs at households that reported smelling poultry odors also had higher odds of CR *E. coli* (OR 1.84, 95% CI 1.21–2.79). Household ownership of food animals did not increase the odds of any outcome among dogs in this study.

Adjusted odds ratios for CR and ESBL-producing *E. coli* carriage in dogs based on household animal care and antibiotic KAP are summarized in Table 5. Adjusted multivariable regression analyses of household animal care and antibiotic KAP found that antibiotic use in any animals at a household within the past 6 months was significantly associated with increased odds of canine ESBL-producing *E. coli* carriage (OR 2.69, 95% CI 1.02–7.10). Specific reasons for antibiotic use, such as to promote animal growth or treat animal illness, were not significantly associated with CR or ESBL-producing *E. coli* carriage in dogs. However, the use of antibiotics based on veterinary/pharmacy recommendation was associated with decreased odds of canine CR *E. coli* carriage (OR 0.18, 95% CI 0.04–0.96). In addition, dogs at

**Table 4. Adjusted odds ratios for CR and ESBL-producing *E. coli* carriage in dogs based on household characteristic risk factors.**

| Risk Factor | CR *E. coli* | | ESBL-producing *E.coli* | |
|---|---|---|---|---|
| | Adjusted OR[1] | 95% CI[2] | Adjusted OR[1] | 95% CI[2] |
| *Caregiver age* | | | | |
| <30 years old (n = 305) | Reference | | | |
| ≥30 years old (n = 250) | 0.82 | 0.55–1.23 | 1.02 | 0.61–1.72 |
| *Household wealth* | | | | |
| Low (n = 166) | Reference | | | |
| Medium/High (n = 389) | 1.02 | 0.69–1.52 | 1.83 | 1.00–3.34 |
| *Household Size* | | | | |
| 1–5 members (n = 433) | Reference | | | |
| >5 members (n = 122) | 1.22 | 0.78–1.91 | 1.15 | 0.63–2.07 |
| *Highest level of caregiver education* | | | | |
| Elementary (n = 164) | Reference | | | |
| High School/University (n = 391) | 1.16 | 0.75–1.77 | 1.29 | 0.71–2.35 |
| *Proximity to nearest commercial food animal facility* | | | | |
| >2 km (n = 117) | Reference | | | |
| 1–2 km (n = 211) | 1.37 | 0.85–2.21 | 1.23 | 0.61–2.48 |
| <1 km (n = 227) | 1.43 | 0.89–2.30 | 1.19 | 0.59–2.37 |
| *Commercial food animal facilities within 5 km* | | | | |
| 0–5 (n = 135) | Reference | | | |
| >5 (n = 419) | **1.72** | **1.15–2.58** | 1.52 | 0.79–2.90 |
| *Commercial poultry odors detected by respondent* | | | | |
| No/don't know (n = 388) | Reference | | | |
| Yes (n = 166) | **1.84** | **1.21–2.79** | 0.98 | 0.57–1.70 |
| *Number of species at household* | | | | |
| 1–3 (n = 422) | Reference | | | |
| >3 (n = 130) | 1.01 | 0.54–1.89 | 1.66 | 0.70–3.94 |
| *Number of animals at household* | | | | |
| 1–5 (n = 312) | Reference | | | |
| 6–20 (n = 131) | 0.79 | 0.38–1.61 | 0.46 | 0.16–1.31 |
| >20 (n = 95) | 0.51 | 0.16–1.64 | 0.70 | 0.12–4.14 |
| *Number of food animals at household* | | | | |
| None (n = 285) | Reference | | | |
| 1–10 (n = 124) | 1.11 | 0.57–2.17 | 2.44 | 0.99–6.02 |
| >10 (n = 129) | 2.11 | 0.69–6.44 | 0.96 | 0.19–4.69 |
| *Livestock units at household[3]* | | | | |
| None (n = 285) | Reference | | | |
| > 0 and ≤ 1 (n = 197) | 1.14 | 0.58–2.24 | 2.42 | 0.98–6.02 |
| >1 (n = 56) | 0.97 | 0.34–2.82 | 1.01 | 0.21–4.75 |
| *Own cat(s)* | | | | |
| No (n = 389) | Reference | | | |
| Yes (n = 149) | 1.09 | 0.70–1.70 | 0.90 | 0.49–1.66 |
| *Own chicken(s)* | | | | |
| No (n = 319) | Reference | | | |
| Yes (n = 219) | 0.69 | 0.36–1.32 | 1.05 | 0.30–3.58 |
| *Own guinea pig(s)* | | | | |
| No (n = 428) | Reference | | | |
| Yes (n = 110) | 1.35 | 0.67–2.73 | 0.47 | 0.17–1.28 |

*(Continued)*

**Table 4.** (*Continued*)

| Risk Factor | CR *E. coli* | | ESBL-producing *E.coli* | |
|---|---|---|---|---|
| | Adjusted OR[1] | 95% CI[2] | Adjusted OR[1] | 95% CI[2] |
| *Own pig(s)* | | | | |
| No (n = 465) | Reference | | | |
| Yes (n = 73) | 0.89 | 0.48–1.63 | 0.87 | 0.37–2.06 |
| *Own rabbit(s)* | | | | |
| No (n = 479) | Reference | | | |
| Yes (n = 59) | 1.01 | 0.53–1.94 | 0.94 | 0.32–2.80 |
| *Own duck(s)* | | | | |
| No (n = 484) | Reference | | | |
| Yes (n = 54) | 1.22 | 0.60–2.47 | 0.57 | 0.18–1.76 |
| *Own cow(s)* | | | | |
| No (n = 497) | Reference | | | |
| Yes (n = 41) | 0.94 | 0.43–2.06 | 0.51 | 0.15–1.76 |

[1]Odds ratio.

[2]95% confidence interval. Bolded numbers indicate statistical significance (α = 0.05).

[3]Livestock units = (0.01) (number of chickens) + (0.30) (number of pigs) + (0.80) (number of cattle) + (0.10) (number of sheep) + (0.10) (number of goats) + (0.02) (number of rabbits) + (0.01) (number of guinea pigs) + (0.03) (number of ducks) + (0.03) (number of quail).

households that obtained antibiotics from a pet food store versus from a veterinarian had increased odds of ESBL-producing *E. coli* (OR 6.83, 95% CI 1.32–35.35). Dogs living at households that reported use of commercial feeds in any of their animals had decreased odds of CR and ESBL-producing *E. coli* carriage (CR OR 0.65, 95% CI 0.43–0.97; ESBL OR 0.50, 95% CI 0.26–0.96). Several similar adjusted odds ratios emerged for 3GCR-MDR and 3GCR-XDR *E. coli* carriage in dogs (S5 and S6 Tables). S3 and S4 Tables show unadjusted odds ratios for CR, ESBL-producing, 3GCR-MDR, and 3GCR-XDR in dogs.

## Risk factors for ceftriaxone-resistant (CR) and ESBL-producing *E. coli* in food animals

Adjusted odds ratios for CR and ESBL-producing *E. coli* carriage in food animals based on household characteristic risk factors are summarized in Table 6. Adjusted odds ratios revealed that food animals at households that owned more than 3 animal species had decreased odds of CR *E. coli* carriage (OR 0.64, 95% CI 0.42–0.97), and food animals at households with more than 5 human members had increased odds of ESBL-producing *E. coli* carriage (OR 2.22, 95% CI 1.23–3.99). Those living at households located within 1 km of the nearest commercial food animal facility also had increased odds of ESBL-producing *E. coli* carriage (OR 2.57, 95% CI 1.08–6.12), though food animals at households that reported smelling poultry odors had lower odds of ESBL-producing *E. coli* carriage (OR 0.48, 95% CI 0.25–0.93). Food animals at households with more than 1 livestock unit also had lower odds of ESBL-producing *E. coli* carriage (OR 0.43, 95% CI 0.19–0.998). Associations between household ownership of specific species and CR and ESBL-producing *E. coli* carriage tended to reflect the prevalence of these AMR phenotypes within each species, as food animals at households that owned chickens, pigs, or ducks had increased odds of CR *E. coli* carriage while those at households with guinea pigs or rabbits had decreased odds of CR *E. coli* carriage (chickens OR 4.92, 95% CI 2.37–10.19; pigs OR 1.52, 95% CI 1.12–2.06; ducks OR 1.87, 95% CI 1.34–2.61; guinea pigs OR 0.58, 95% CI 0.38–0.88; rabbits OR 0.65, 95% CI 0.47–0.90).

**Table 5. Adjusted odds ratios for CR and ESBL-producing *E. coli* carriage in dogs based on household animal care and antibiotic knowledge, attitudes, and practices (KAP) risk factors.**

| Risk Factor | CR *E. coli* | | ESBL-producing *E. coli* | |
|---|---|---|---|---|
| | Adjusted OR[1] | 95% CI[2] | Adjusted OR[1] | 95% CI[2] |
| *Antibiotics given to any animals in past 6 months* | | | | |
| No (n = 467) | Reference | | | |
| Yes (n = 71) | 3.31 | 0.95–11.58 | **2.69** | **1.02–7.10** |
| *Antibiotics given to dogs in past 6 months* | | | | |
| No (n = 510) | Reference | | | |
| Yes (n = 28) | 2.41 | 0.92–6.32 | 1.78 | 0.32–9.89 |
| *Other medications/vitamins given in past 6 months* | | | | |
| No (n = 440) | Reference | | | |
| Yes (n = 96) | 1.03 | 0.63–1.68 | 0.79 | 0.36–1.71 |
| *Use antibiotics for growth promotion*[3] | | | | |
| No (n = 44) | Reference | | | |
| Yes (n = 21) | 0.66 | 0.21–2.05 | 1.52 | 0.12–18.62 |
| *Use antibiotics for animal illness prevention*[3] | | | | |
| No (n = 49) | Reference | | | |
| Yes (n = 17) | 0.87 | 0.19–4.06 | 0.84 | 0.08–9.00 |
| *Use antibiotics for animal illness treatment*[3] | | | | |
| No (n = 55) | Reference | | | |
| Yes (n = 9) | 5.29 | 0.48–58.95 | 0.53 | 0.02–12.30 |
| *Use antibiotics based on veterinary/ pharmacy recommendation*[3] | | | | |
| No (n = 55) | Reference | | | |
| Yes (n = 11) | **0.18** | **0.04–0.96** | 0.59 | 0.10–3.55 |
| *Antibiotic Source*[3] | | | | |
| Veterinarian (n = 47) | | | Reference | |
| Pet food store (n = 17) | 2.82 | 0.66–12.06 | **6.83** | **1.32–35.35** |
| *Veterinary access* | | | | |
| No (n = 490) | Reference | | | |
| Yes (n = 48) | 0.67 | 0.36–1.24 | 0.41 | 0.12–1.41 |
| *Animals consumed river or irrigation water in past 3 weeks* | | | | |
| No (n = 508) | Reference | | | |
| Yes (n = 44) | 1.32 | 0.65–2.66 | 0.25 | 0.05–1.23 |
| *Animals fed commercial feed* | | | | |
| No/Don't know (n = 348) | Reference | | | |
| Yes (n = 184) | **0.65** | **0.43–0.97** | **0.50** | **0.26–0.96** |
| *Household member slaughtered livestock/poultry, worked with animals, or worked in animal or animal by-product processing in past 6 months* | | | | |
| No/Don't know (n = 413) | Reference | | | |
| Yes (n = 142) | 1.46 | 0.94–2.25 | 1.34 | 0.77–2.32 |
| *Household member worked with animal or human feces outside the home in past 6 months* | | | | |
| No/Don't know (n = 494) | Reference | | | |
| Yes (n = 61) | 1.01 | 0.56–1.80 | 0.98 | 0.42–2.28 |
| *Household member took antibiotics in past 3 months*[4] | | | | |
| No (n = 33) | Reference | | | |
| Yes (n = 138) | 0.92 | 0.39–2.17 | 0.48 | 0.16–1.41 |
| *Animals allowed inside the home* | | | | |
| No/Don't know (n = 286) | Reference | | | |
| Yes (n = 263) | 1.20 | 0.84–1.73 | 0.94 | 0.57–1.57 |

*(Continued)*

**Table 5.** (Continued)

| Risk Factor | CR *E. coli* | | ESBL-producing *E. coli* | |
|---|---|---|---|---|
| | Adjusted OR[1] | 95% CI[2] | Adjusted OR[1] | 95% CI[2] |
| *Animals allowed near children* | | | | |
| No/Don't know (n = 216) | Reference | | | |
| Yes (n = 334) | 1.24 | 0.86–1.79 | 1.14 | 0.68–1.92 |
| *Animal feces management* | | | | |
| Place in trash (n = 264) | Reference | | | |
| Leave in yard (n = 99) | 0.99 | 0.60–1.63 | 1.56 | 0.80–3.06 |
| Store and place on land/ Use as fertilizer (n = 133) | 1.11 | 0.65–1.87 | 0.71 | 0.30–1.68 |
| *Can antibiotics kill bacteria?* | | | | |
| "Yes"/Correct (n = 210) | Reference | | | |
| "No"/Incorrect (n = 113) | 0.98 | 0.60–1.61 | 0.98 | 0.49–1.99 |
| Don't know (n = 232) | 1.09 | 0.73–1.64 | 1.21 | 0.68–2.14 |
| *Can antibiotics kill viruses?* | | | | |
| "No"/Correct (n = 116) | Reference | | | |
| "Yes"/Incorrect (n = 207) | 0.92 | 0.56–1.51 | 1.24 | 0.60–2.54 |
| Don't know (n = 231) | 1.04 | 0.64–1.70 | 1.41 | 0.69–2.87 |

[1]Odds ratio.

[2]95% confidence interval. Bolded numbers indicate statistical significance (α = 0.05).

[3]Questions regarding motivation for antibiotic use and antibiotic source were only answered by those caregivers that reported using antibiotics for their animal(s).

[4]Household member use of antibiotics was determined based on caregiver response to whether or not their child in the study had taken antibiotics in the past 3 months and whether or not a household member had taken antibiotics in the past 3 months, the latter of which was only asked to those who reported having a household member with an illness or infection in the past 3 months.

Adjusted odds ratios for CR and ESBL-producing *E. coli* carriage in food animals based on household animal care and antibiotic KAP are summarized in Table 7. The majority of household animal care and antibiotic KAP risk factors were not significantly associated with food animal carriage of CR or ESBL-producing *E. coli*. Neither antibiotic use in any animals at the household nor use specifically in food animals yielded significant odds ratios for food animal CR or ESBL-producing *E. coli* carriage. However, the use of antibiotics to promote growth (OR 0.41, 95% CI 0.19–0.89) and the purchase of antibiotics from a pet food store (OR 0.47, 95% CI 0.25–0.89) resulted in lower odds of food animal CR *E. coli* carriage. Adjusted odds ratios for risk factors for 3GCR-MDR and 3GCR-XDR *E. coli* carriage in food animals are summarized in S9 and S10 Tables. S7 and S8 Tables show unadjusted odds ratios for CR, ESBL-producing, 3GCR-MDR, and 3GCR-XDR *E. coli* carriage in food animals.

## Discussion

This study found a high prevalence of CR and ESBL-producing *E. coli* among fecal samples from domestic animals at households in semirural parishes east of Quito, Ecuador. Prevalence varied considerably by species, with all AMR outcomes most common in dogs, pigs, chickens, and ducks, suggesting these species warrant further attention for their roles in ESBL-producing and other AR *E. coli* transmission. The prevalence of CR *E. coli* among all animal fecal samples collected during this study was 56%, while that of ESBL-producing *E. coli* was 10%. Interestingly, this ESBL prevalence is consistent with the estimated prevalence of ESBL-producing *E. coli* intestinal carriage among humans in the Americas [3].

**Table 6. Adjusted odds ratios for CR and ESBL-producing *E. coli* carriage in food animals based on household characteristic risk factors.**

| Risk Factor | CR *E. coli* | | ESBL-producing *E.coli* | |
|---|---|---|---|---|
| | Adjusted OR[1] | 95% CI[2] | Adjusted OR[1] | 95% CI[2] |
| *Caregiver age* | | | | |
| <30 years old (n = 354) | Reference | | | |
| ≥30 years old (n = 283) | 1.33 | 0.95–1.86 | 0.54 | 0.26–1.13 |
| *Household wealth* | | | | |
| Low (n = 222) | Reference | | | |
| Medium/High (n = 415) | 0.84 | 0.62–1.14 | 0.82 | 0.45–1.51 |
| *Household Size* | | | | |
| 1–5 members (n = 480) | Reference | | | |
| >5 members (n = 157) | 1.06 | 0.76–1.48 | **2.22** | **1.23–3.99** |
| *Highest level of caregiver education* | | | | |
| Elementary (n = 211) | Reference | | | |
| High School/University (n = 426) | 1.26 | 0.88–1.81 | 0.53 | 0.26–1.08 |
| *Proximity to nearest commercial food animal facility* | | | | |
| >2 km (n = 159) | Reference | | | |
| 1–2 km (n = 197) | 0.83 | 0.55–1.24 | 1.89 | 0.76–4.68 |
| <1 km (n = 281) | 0.97 | 0.65–1.45 | **2.57** | **1.08–6.12** |
| *Commercial food animal facilities within 5 km* | | | | |
| 0–5 (n = 134) | Reference | | | |
| >5 (n = 503) | 1.09 | 0.77–1.54 | 1.31 | 0.63–2.69 |
| *Commercial poultry odors detected by respondent* | | | | |
| No/don't know (n = 395) | Reference | | | |
| Yes (n = 242) | 0.86 | 0.63–1.18 | **0.48** | **0.25–0.93** |
| *Number of species at household* | | | | |
| 1–3 (n = 189) | Reference | | | |
| >3 (n = 444) | **0.64** | **0.42–0.97** | 1.06 | 0.48–2.35 |
| *Number of animals at household* | | | | |
| 1–5 (n = 53) | Reference | | | |
| 6–20 (n = 214) | 1.05 | 0.55–1.99 | 0.66 | 0.21–2.04 |
| >20 (n = 342) | 0.80 | 0.32–2.01 | 0.75 | 0.10–5.66 |
| *Number of food animals at household* | | | | |
| ≤ 10 (n = 183) | Reference | | | |
| 11–20 (n = 103) | 1.00 | 0.61–1.62 | 1.09 | 0.43–2.72 |
| >20 (n = 323) | 1.50 | 0.71–3.16 | 0.86 | 0.15–5.01 |
| *Livestock units at household[3]* | | | | |
| ≤ 1 (n = 355) | Reference | | | |
| >1 (n = 254) | 1.24 | 0.87–1.77 | **0.43** | **0.19–0.998** |
| *Own dog(s)* | | | | |
| No (n = 48) | Reference | | | |
| Yes (n = 561) | 0.94 | 0.52–1.69 | 0.59 | 0.22–1.59 |
| *Own cat(s)* | | | | |
| No (n = 331) | Reference | | | |
| Yes (n = 278) | 1.27 | 0.94–1.72 | 1.60 | 0.85–3.03 |
| *Own chicken(s)* | | | | |
| No (n = 63) | Reference | | | |
| Yes (n = 546) | **4.92** | **2.37–10.19** | 1.30 | 0.48–3.50 |
| *Own guinea pig(s)* | | | | |

(*Continued*)

**Table 6.** (Continued)

| Risk Factor | CR *E. coli* | | ESBL-producing *E.coli* | |
|---|---|---|---|---|
| | Adjusted OR[1] | 95% CI[2] | Adjusted OR[1] | 95% CI[2] |
| No (n = 236) | Reference | | | |
| Yes (n = 373) | **0.58** | **0.38–0.88** | 0.67 | 0.34–1.34 |
| *Own pig(s)* | | | | |
| No (n = 333) | Reference | | | |
| Yes (n = 276) | **1.52** | **1.12–2.06** | 1.24 | 0.61–2.51 |
| *Own rabbit(s)* | | | | |
| No (n = 387) | Reference | | | |
| Yes (n = 222) | **0.65** | **0.47–0.90** | 0.60 | 0.29–1.23 |
| *Own duck(s)* | | | | |
| No (n = 409) | Reference | | | |
| Yes (n = 200) | **1.87** | **1.34–2.61** | 1.18 | 0.61–2.28 |
| *Own cow(s)* | | | | |
| No (n = 416) | Reference | | | |
| Yes (n = 193) | 1.38 | 0.96–1.97 | 0.75 | 0.32–1.72 |

[1]Odds ratio.

[2]95% confidence interval. Bolded numbers indicate statistical significance ($\alpha = 0.05$).

[3]Livestock units = (0.01) (number of chickens) + (0.30) (number of pigs) + (0.80) (number of cattle) + (0.10) (number of sheep) + (0.10) (number of goats) + (0.02) (number of rabbits) + (0.01) (number of guinea pigs) + (0.03) (number of ducks) + (0.03) (number of quail).

Most research assessing AR *E. coli* prevalence among domestic animals in Ecuador has focused on chickens and dogs. Information on the prevalence among other animal species is limited. Surveillance of AR *E. coli* in small-scale and backyard poultry production in remote communities in Esmeraldas Province, Ecuador has identified resistance to cefotaxime, a third-generation cephalosporin, in as many as 66.1% of farmed broiler chickens and 17.9% of backyard chickens not fed antimicrobials [24]. In Esmeraldas, community exposure to broiler chicken production resulted in possible spillover into backyard chickens, leading to increased cefotaxime resistance in backyard chickens independent of antimicrobial use or direct contact with broiler poultry [24]. Multidrug resistance to amoxicillin/clavulanic acid, cephalothin, cefotaxime, and gentamicin has also been found in small-scale production settings in Esmeraldas exclusively in birds raised for commercial purposes (versus in backyard/household flocks raised for domestic use). These antimicrobials have thus been referred to as "production bird signatures" [56]. Among industrial poultry facilities in Quito, third-generation cephalosporin resistance has been documented, and was recently identified in 91.7% of commercial poultry cecal samples in one study [77].

Here, we identified CR and ESBL-producing *E. coli* in 70% and 12% of all chicken fecal samples. While the amount of third-generation cephalosporin resistance observed in poultry in this study was generally less than that observed in industrial facilities, it was greater than that previously observed in backyard chickens in Esmeraldas Province [56, 77, 78]. Furthermore, 3GCR-MDR strains resistant to "production bird signature" antimicrobials (with cefepime substituted for cephalothin, both first-generation cephalosporins) were identified in 18 isolates in this study, including in 5 chicken isolates [56]. This study did not distinguish between small-scale broiler and backyard chickens, but the presence of "production bird signature" AMR phenotypes in chickens in this study could represent spillover from production poultry. Further data would be needed to determine the prevalence of this signature pattern in

**Table 7. Adjusted odds ratios for CR and ESBL-producing *E. coli* carriage in food animals based on household animal care and antibiotic knowledge, attitudes, and practices (KAP) risk factors.**

| Risk Factor | CR *E. coli* | | ESBL-producing *E. coli* | |
|---|---|---|---|---|
| | Adjusted OR[1] | 95% CI[2] | Adjusted OR[1] | 95% CI[2] |
| *Antibiotics given to any animals in past 6 months* | | | | |
| No (n = 482) | Reference | | | |
| Yes (n = 127) | 0.67 | 0.44–1.01 | 0.67 | 0.16–2.73 |
| *Antibiotics given to livestock/poultry in past 6 months* | | | | |
| No (n = 540) | | | | |
| Yes (n = 67) | 0.76 | 0.47–1.21 | 0.34 | 0.08–1.46 |
| *Other medications/vitamins given in past 6 months* | | | | |
| No (n = 486) | Reference | | | |
| Yes (n = 122) | 1.23 | 0.83–1.83 | 1.30 | 0.61–2.79 |
| *Use antibiotics for growth promotion*[3] | | | | |
| No (n = 77) | Reference | | | |
| Yes (n = 40) | **0.41** | **0.19–0.89** | 0.27 | 0.02–4.95 |
| *Use antibiotics for animal illness prevention*[3] | | | | |
| No (n = 87) | Reference | | | |
| Yes (n = 30) | 0.85 | 0.39–1.84 | 1.69 | 0.37–7.76 |
| *Use antibiotics for animal illness treatment*[3] | | | | |
| No (n = 93) | Reference | | | |
| Yes (n = 19) | 1.90 | 0.33–10.99 | - | - |
| *Use antibiotics based on veterinary/ pharmacy recommendation*[3] | | | | |
| No (n = 104) | Reference | | | |
| Yes (n = 13) | 0.29 | 0.07–1.14 | - | - |
| *Antibiotic Source*[3] | | | | |
| Veterinarian (n = 78) | Reference | | | |
| Pet food store (n = 46) | **0.47** | **0.25–0.89** | 1.36 | 0.23–8.18 |
| *Veterinary access* | | | | |
| No (n = 532) | Reference | | | |
| Yes (n = 77) | 1.46 | 0.91–2.33 | 0.70 | 0.24–2.03 |
| *Animals consumed river or irrigation water in past 3 weeks* | | | | |
| No (n = 500) | Reference | | | |
| Yes (n = 133) | 0.85 | 0.59–1.21 | 0.50 | 0.19–1.33 |
| *Animals fed commercial feed* | | | | |
| No/Don't know (n = 296) | Reference | | | |
| Yes (n = 302) | 1.21 | 0.89–1.64 | 0.81 | 0.42–1.56 |
| *Household member slaughtered livestock/poultry, worked with animals, or worked in animal or animal by-product processing in past 6 months* | | | | |
| No/Don't know (n = 376) | Reference | | | |
| Yes (n = 261) | 1.33 | 1.00–1.76 | 1.38 | 0.78–2.44 |
| *Household member worked with animal or human feces outside the home in past 6 months* | | | | |
| No/Don't know (n = 534) | Reference | | | |
| Yes (n = 103) | 0.93 | 0.62–1.38 | 0.46 | 0.17–1.27 |
| *Household member took antibiotics in past 3 months*[4] | | | | |
| No (n = 26) | Reference | | | |
| Yes (n = 171) | 0.75 | 0.40–1.41 | 0.49 | 0.15–1.57 |
| *Animals allowed inside the home* | | | | |
| No/Don't know (n = 360) | Reference | | | |
| Yes (n = 272) | 1.04 | 0.78–1.39 | 0.63 | 0.33–1.18 |

*(Continued)*

**Table 7.** (Continued)

| Risk Factor | CR *E. coli* | | ESBL-producing *E. coli* | |
|---|---|---|---|---|
| | Adjusted OR[1] | 95% CI[2] | Adjusted OR[1] | 95% CI[2] |
| *Animals allowed near children* | | | | |
| No/Don't know (n = 246) | Reference | | | |
| Yes (n = 387) | 0.90 | 0.67–1.23 | 0.56 | 0.31–1.01 |
| *Animal feces management* | | | | |
| Place in trash (n = 143) | Reference | | | |
| Leave in yard (n = 136) | 1.29 | 0.79–2.10 | 0.91 | 0.40–2.06 |
| Store and place on land/ Use as fertilizer (n = 313) | 1.25 | 0.85–1.83 | 1.25 | 0.59–2.64 |
| *Can antibiotics kill bacteria?* | | | | |
| "Yes"/Correct (n = 242) | Reference | | | |
| "No"/Incorrect (n = 100) | 0.71 | 0.46–1.09 | 1.48 | 0.70–3.14 |
| Don't know (n = 291) | 0.83 | 0.60–1.15 | 1.00 | 0.50–1.98 |
| *Can antibiotics kill viruses?* | | | | |
| "No"/Correct (n = 115) | Reference | | | |
| "Yes"/Incorrect (n = 220) | 1.13 | 0.73–1.75 | 0.82 | 0.38–1.80 |
| Don't know (n = 300) | 1.03 | 0.67–1.57 | 0.85 | 0.39–1.86 |

[1]Odds ratio. A (-) indicates a positivity violation prevented odds ratio (OR) calculation.

[2]95% confidence interval. Bolded numbers indicate statistical significance (α = 0.05).

[3]Questions regarding motivation for antibiotic use and antibiotic source were only answered by those caregivers that reported using antibiotics for their animal(s).

[4]Household member use of antibiotics was determined based on caregiver response to whether or not their child in the study had taken antibiotics in the past 3 months and whether or not a household member had taken antibiotics in the past 3 months, the latter of which was only asked to those who reported having a household member with an illness or infection in the past 3 months.

commercial poultry in the Quito region. The substantial prevalence of AMR among chickens and other poultry in this study confirms the importance of continued AMR surveillance in both large and small-scale poultry ownership settings [23]. Future efforts should focus on the role backyard and small-scale poultry production may play in promoting the AMR prevalence observed here, as well as potential pathways for spillover in this region.

Previously, MDR *E. coli* was isolated from 40% of canine fecal samples collected in a Quito park, and ESBL-producing *E. coli* was common amongst these samples, highlighting the need for AMR surveillance of canine feces in public settings [41]. Globally, the prevalence of ESBL-producing *E. coli* among dogs is estimated to be approximately 6.87% [79]. We found CR, ESBL-producing, and 3GCR-MDR *E. coli* in 68%, 13%, and 63% of all canine fecal samples, respectively, collected over three sampling cycles. While any comparison of prevalence must be done with caution given variations in sampling, susceptibility testing, and statistical analyses, the increased prevalence of ESBL and 3GCR-MDR *E. coli* seen here could be due in part to factors such as the semi-rural setting of this study, varying antimicrobial exposures between study sites, and differences in fecal collection practices by owners in public versus private settings. Additionally, our data do not represent the true prevalence of 3GCR-MDR *E. coli* because we isolated *E. coli* in a medium containing ceftriaxone. Our findings confirm the importance of AMR surveillance among dogs in both public and private settings and emphasize the need for greater focus on the role dogs may play in AMR transmission in the Quito region and beyond.

Shared ESBL-producing *E. coli* isolates, AMR genes, and AMR replicons have been identified in pets, food animals, and children within the study region [13, 61]. With ESBL and other

AMR genes circulating among *E. coli* in domestic animals in this community, the risk of spill-over into human populations, including possible gene or plasmid transfer to pathogenic (mostly opportunistic) *E. coli* strains and subsequent human infection, is a concern [13, 23, 61]. Transfer of pathogenic *E. coli* clones among humans and dogs has been documented elsewhere, including among human and canine members of one household [80]. Species with a high prevalence of AR *E. coli* carriage that are more likely to range freely, such as dogs, chickens, and ducks, may present heightened risk for transmission to humans, as free-ranging animals may have a higher likelihood of exposure to AR bacteria outside the household environment and subsequent direct contact with humans, as well as more widespread contribution to environmental contamination [61, 81–83]. However, further research about the dominant mechanisms and pathways of AMR transmission in this community are needed prior to drawing firm conclusions.

In this study, we identified several potential risk factors for dog and food animal colonization with CR and ESBL-producing *E. coli* in households of semi-rural parishes east of Quito, Ecuador. Elsewhere, proximity to other food animal facilities has been implicated in the risk of AMR in food animals [25, 45]. Commercial food animal production is often linked to AMR not only because of antimicrobial use in such contexts but also because of conditions that may promote AMR, such as overcrowding and poor sanitation practices [84, 85]. In our study, dogs that lived at households that reported smelling poultry odors and that lived within 5 km of more than five commercial food animal facilities had higher odds of CR and 3GCR-MDR *E. coli* carriage. Food animals also had increased odds of ESBL-producing *E. coli* colonization when living within 1 km of the nearest commercial food animal facility. Together, these findings suggest that commercial food animal facilities may play a role in domestic animal CR and ESBL-producing *E. coli* colonization in this setting as well.

AR *E. coli* is likely highly prevalent on local commercial poultry farms [77]. Domestic animals located closer to commercial food animal facilities, particularly those more mobile species such as dogs, chickens, and other poultry, may be more likely to be exposed to AR organisms and genes from commercial facilities through increased exposure to contaminated environments [61, 86]. Practices such as the use of commercial poultry feces as fertilizer or contamination of shared water sources by such facilities could contribute to these results [87]. There is limited information regarding the prevalence of AMR in environmental water sources and the role commercial livestock play in contributing to such sources of AMR throughout Ecuador, including in the Pichincha province, where the majority of rivers sampled in one study had coliform units above Ecuadorian guidelines [88, 89]. A better understanding of the environmental AR *E. coli* reservoirs in this region would help to better contextualize these findings.

Interestingly, proximity to closest food animal operation did not significantly alter the odds ratio of ESBL-producing *E. coli* carriage in dogs as it did for food animals. In northwestern Ecuador, distance to closest small-scale broiler chicken farming operation was not associated with AR *E. coli* isolation from humans or chickens, and the highly mobile nature of backyard chickens, found to travel 0–59 meters away from their households, was implicated in this lack of a correlation [86]. In our study region, dogs often have similar liberty to roam freely [61]. Canine exposure to environments contaminated by commercial food animal facilities through such mobility, which may be more impacted by the density of such facilities in the household area rather than distance to the closest facility, could explain the results seen here.

Despite the observed connection between proximity to commercial food animal facilities and food animal ESBL-producing *E. coli* colonization, food animals at households that reported smelling poultry odors had decreased odds of ESBL-producing *E. coli* carriage. Smelling poultry odors is a crude and subjective measure of proximity to and density of food animal operations and could reflect variables such as wind patterns and when household members are

most likely to be home and notice such odors. This result should thus be interpreted with caution. Overall, these findings highlight the need for greater understanding of the role that commercial food animal facilities play in AR *E. coli* transmission among domestic animals in this region, and the potential pathways through which this transmission may occur.

In other settings, a dog's direct contact with livestock has been found to increase odds of canine ESBL-producing *E. coli* carriage, and companion animal presence on farms in Madagascar has been found to increase odds of ESBL-producing *E. coli* in beef cattle, likely due to the mobility of dogs around properties in these settings [28, 47]. We did not find increased odds of CR or ESBL-producing *E. coli* amongst dogs living at households with food animals in this study. Similarly, neither dog nor cat ownership was significantly associated with any AMR outcome in food animals. While a household's ownership of food animals may be a good indicator of a dog's exposure to these animals, it is possible this variable overlooks the intricacies of such exposures. Dogs living at households without food animals may still be exposed to such animals through frequenting other properties where food animals are housed, consuming raw meat as part of their regular diet, scavenging, or through exposure to food animal fecal contamination in water and other environmental sources [28, 39]. The impact of companion and food animal interaction on AMR transmission in this study's regions thus necessitates further exploration, and is likely more complex than simple exposure within the household environment.

Unsurprisingly, dogs previously treated with antibiotics had increased odds of 3GCR-MDR *E. coli*, consistent with findings in clinical settings [47, 49]. The odds of canine carriage of ESBL-producing *E. coli*, in addition to 3GCR-MDR, was also associated with the use of antibiotics in any household animal. These findings indicate that dogs might be exposed to *E. coli* carrying ESBL-producing and other AMR genes in their household environments not only when they are treated with antibiotics, but also when other animals on the premises consume these drugs. However, antibiotic use in domestic animals was not associated with increased odds of CR, ESBL-producing, 3GCR-MDR, or 3GCR-XDR *E. coli* colonization in food animals in this study. Previous research in large and small-scale settings have found that antimicrobial use and specific treatment regimens are not always directly correlated with levels of AMR observed in food animals [25, 29, 35]. In other contexts, specific sociocultural livelihood factors that promote bacterial transmission, such as animal movement and integration practices, may be more important than the use of antimicrobials themselves in elevating AMR risk [35]. Though ESBL and other AMR genes have historically been thought to confer bacterial fitness costs, this paradigm appears to be shifting, and thus these genes may persist even in the absence of antimicrobial selective pressures [90–92]. The difference between antibiotic use risk factors for dogs and food animals observed in our study highlights the complexity of addressing AMR. With ESBL and other AMR genes widespread among people, animals, and the environment, factors beyond antimicrobial stewardship alone are likely important in addressing this problem effectively.

In other small-scale poultry contexts, use of commercial feeds has been found to increase the odds of MDR *E. coli* carriage in chickens [26]. In this study region, commercial feed is most often used for food animals, though some producers in this setting have reported use of commercial feed in dogs as well [60]. Many commercial feeds have historically contained antimicrobials to promote growth and prevent illness, and many are still presumed to do so [23, 26, 56, 93–95]. However, though antimicrobials have been identified in commercial poultry feeds in northwestern Ecuador [56], a previous review of commonly used commercial feeds in semirural parishes outside of Quito found that no commonly used commercial feed brands in the area contained antimicrobials [60]. This fact might explain the lack of an association between commercial feed use and any AMR outcomes in food animals seen here.

Unexpectedly, the odds of CR and ESBL-producing *E. coli* carriage among dogs were lower in households that reported commercial feed use in any animals. This relationship between canine ESBL-producing *E. coli* colonization and commercial feed use could be influenced by the fact that dogs consuming commercial feed may be less likely to consume raw meat or poultry, both of which have been implicated in increasing ESBL-producing *E. coli* risk [39, 43]. Elsewhere, dogs that consume dry food have also been found to have decreased odds of ESBL-producing *E. coli* [39]. Complicating analysis of the results here, the survey used in this study did not ask owners to specify which animals receive commercial feed. An analysis of commonly used commercial feeds specifically in dogs and their antimicrobial components in this setting would be helpful in elucidating the relationship observed here.

Food animals at households with more than 5 people also had increased odds of ESBL-producing *E. coli* carriage in this study. We have not observed previous reports of such associations in food animals, though in humans, AMR has been linked to household crowding [96], and crowding of 2.5–8 people per room in this study's region has been previously weakly associated with increased though insignificant odds of ESBL-producing, 3GCR-MDR, and 3GCR-XDR *E. coli* in humans [62]. Similar pathways related to household crowding, as well as the potential for an increased number of food animal and human exposures in households with more people, may drive some fraction of the resistance observed in both food animals and humans in this setting.

AMR has been associated with overcrowding in animal housing in both large and small scale settings [26, 84, 85, 97], but we found decreased odds of ESBL-producing *E. coli* colonization among food animals living at households with more than one livestock unit. Furthermore, food animals at households with more than three total species had decreased odds of CR *E. coli* carriage. This surprising relationship necessitates further exploration. Our findings might be explained by differences in mechanisms by which animals are housed when more species and/ or more than one livestock unit is present at the household. For example, households with a greater number of species or livestock units may be more likely to separate animals in different ways, minimize free-roaming, or practice different water, sanitation, and hygiene (WaSH) behaviors than those with fewer livestock units. Assessing such practices would be important in interpreting these results.

In other contexts, ESBL and other AR *E. coli* have also been connected to specific cleaning and sanitation regimens, sanitation practices during and after interaction with animals, rodent, fly, or other vector control, animal movement to and from households, and animal housing practices [27, 28, 31, 35, 45, 98, 99]. In this study, animal exposure to children in and around the home did not appear to alter the odds of any AMR outcome of interest in dogs or food animals, but we did not collect information regarding water, sanitation, and hygiene (WaSH) practices as they pertain to the caretaker-animal interactions at each household. A better understanding of caregiver-animal interactions, including sanitation, handling, and other practices, could help us to better understand which exposures and management practices pose highest risk.

Several findings in this study suggest that the role of veterinarians in antimicrobial prescribing and AMR mitigation warrants closer attention. Sales agents at veterinary supply stores in this region, including veterinarians, cashiers, and store owners, frequently recommend an inappropriate antimicrobial class for disease treatment and/or the use of antimicrobials to promote animal growth [57]. There is room for increased support and oversight in this realm, as such veterinary sales agents may be an effective target for upstream drivers of AMR in this setting [57]. Multiple veterinary-related risk factors were found to significantly alter the odds of AR *E. coli* carriage among animals in this study, suggesting that antimicrobial prescription or recommendation practices among veterinarians and sales agents could be an important target

for addressing upstream drivers of AMR in this region. However, working to expand such antimicrobial stewardship requires an understanding of the challenges that impede appropriate antimicrobial prescription. Factors such as workload, economic considerations, discomfort challenging older colleagues, and individual values influence veterinary prescribing practices, and recognition of such factors is important in increasing antimicrobial stewardship [100–105]. Furthermore, it is important to note that the majority of participants in this study did not have access to veterinary care. While improving veterinary and sales agent oversight is important, such efforts should occur alongside efforts to expand such veterinary care. A focus on addressing AMR risk factors beyond veterinary care will also be crucial in equitable intervention design.

This study also found evidence that the ways in which veterinary involvement affects AMR may differ among species. We found that food animals at households that purchased antibiotics from a pet food store, rather than from a veterinarian, had decreased odds of CR *E. coli* carriage. Food animals at households with access to veterinary care also had increased odds of 3GCR-XDR *E. coli* carriage. Surprisingly, food animals at households that used antibiotics for growth promotion also had lower odds of CR *E. coli* carriage, which could be related to specific pet food store recommendation practices that encourage antibiotic use for such growth promotion or decreased veterinary involvement when such practices are used. In contrast, dogs at households that obtained antibiotics from a pet food store, rather than from a veterinarian, had increased odds of ESBL-producing *E. coli* colonization. The ways in which veterinary access impacts dog versus food animal AR *E. coli* carriage in this region is therefore nuanced. The regimens of antimicrobial treatment prescribed by veterinarians and recommended by pet food stores most likely differs depending on the species and prescriber knowledge. For example, a veterinarian more comfortable with companion animals may be more likely to advise antimicrobial stewardship and appropriate antimicrobial selection among dogs but less likely to do so among food animals. Veterinary support may additionally be relied upon more often by small-scale producers for certain species rather than others, further altering the ways in which veterinary access would impact AR *E. coli* carriage among varying species. Caregiver experience may also contribute to these differences, as more experienced small-scale producers may be less likely to consult with a veterinarian and more likely to select the appropriate antimicrobials from a pet food store. The intricacies of how, when, and which veterinarians are involved in small-scale food animal production in this setting must therefore be better understood to determine best intervention strategies.

Efforts to address AMR moving forward must also be balanced with an emphasis on protecting food security and accommodating specific socioeconomic and cultural contexts that could influence effective intervention design [22, 35]. The diverse risk factors identified in this study highlight the complexity of addressing AMR and the need to understand local contexts. Mitigation and control require both larger-reaching policy as well as targeted interventions relevant to local settings. Collaboration between all stakeholders involved will be key in future efforts to address this global health threat.

This study had several important limitations. Household loss to follow-up occurred when household caregivers moved, elected to unenroll, or were otherwise unavailable for sample collection. We attempted to address potential selection bias due to loss to follow-up both by enrolling new households that met the inclusion criteria when previous households were lost and by using statistical methods (GEE) that can account for imbalanced data. Given the tendency for cats to defecate away from the household environment, feline sample discovery was difficult. Therefore, risk factors for feline carriage of CR and ESBL-producing *E. coli* could not be determined in this study despite the common occurrence of feline ownership. While this does not alter the results obtained for canines and food animal fecal samples in this study, it is

likely the risk factors for feline CR and ESBL-producing *E. coli* carriage varies from the species explored here, and a different study design would be needed to better understand these risk factors. As few participants reported antibiotic use in their animals, we were also unable to calculate odds ratios for some risk factors related to antibiotic use due to positivity violations. An expanded study population would assist in this challenge and increase our ability to better identify differences in risk factors among species. In addition, all surveys relied on self-report from study participants, potentially introducing reporting and/or interviewer bias. Participants may have misremembered, not known certain information requested, been hesitant to offer information about antimicrobial use, or been more likely to offer socially acceptable answers. An assessment of risk at the individual household level also only tells one part of the story; as animals may be exposed to other risk factors throughout the community, certain exposures may have been misclassified in this analysis. We were not able to adjust for clustering and did not adjust for multiple comparisons, as this was an exploratory analysis to identify potential risk factors for AMR in domestic animals warranting future investigation. Future analyses may include the assessment of risk factors at the community level using hierarchical modeling methods or adjustment for such clustering. Further research is also needed to identify CR, ESBL-producing, 3GCR-MDR, and 3GCR-XDR *E. coli* environmental pathways in this study area to directly characterize exposure routes among household animals.

## Conclusions

This study identified a high prevalence of CR, ESBL-producing, 3GCR-MDR, and 3GCR-XDR *E. coli* among domestic animals of households in semirural parishes east of Quito, Ecuador, particularly among dogs, pigs, chickens, and ducks. Risk factors contributing to canine and food animal colonization with CR and ESBL-producing *E. coli*, such as commercial food animal facility exposure, antimicrobial use, and veterinary involvement, were varied and complex, highlighting the context-dependent and multifaceted approach necessary to address AMR more broadly. Future studies assessing specific mechanisms of transmission that occur between animals, humans, and the environment would help to further elucidate the role of domestic animals in AMR transmission in the region, allowing more focused and evidence-based mitigation and control strategies. Any efforts to address AMR here or elsewhere would benefit from taking such a One Health approach.

## Supporting information

**S1 Survey. Survey template used in this study (Spanish).**
(PDF)

**S2 Survey. Survey template used in this study (English translation).**
(PDF)

**S1 Table. Adjusted logistic regression models, including confounders, for risk factors of interest.**
(PDF)

**S2 Table. *E. coli* antimicrobial resistance patterns by species and cycle[1].** [1]Fecal samples from "other" species, including llamas (2), cats (1), and hamsters (1), produced no CR *E. coli* isolates and so were not included in this table. [2]Other poultry = geese, pigeon, and quail. [3]Percentage of ceftriaxone-resistant (CR) *E. coli* isolates. [4]3GCR-MDR and 3GCR-XDR *E. coli* were determined from isolates resistant to ceftriaxone.
(PDF)

**S3 Table. Unadjusted odd ratios for CR, ESBL-producing, 3GCR-MDR, and 3GCR-XDR *E. coli* carriage in dogs based on household risk factors.** [1]3GCR-MDR and 3GCR-XDR *E. coli* were determined from isolates resistant to ceftriaxone. [2]Odds ratio. [3]95% confidence interval. Bolded numbers indicate statistical significance (α = 0.05). [4]Livestock units = (0.01) (number of chickens) + (0.30) (number of pigs) + (0.80) (number of cattle) + (0.10) (number of sheep) + (0.10) (number of goats) + (0.02) (number of rabbits) + (0.01) (number of guinea pigs) + (0.03) (number of ducks) + (0.03) (number of quail).
(PDF)

**S4 Table. Unadjusted odds ratios for CR, ESBL-producing, 3GCR-MDR, and 3GCR-XDR *E. coli* carriage in dogs based on household animal care and antibiotic knowledge, attitudes, and practices (KAP) risk factors.** [1] 3GCR-MDR and 3GCR-XDR *E. coli* were determined from isolates resistant to ceftriaxone. [2] Odds ratio. [3] 95% confidence interval. Bolded numbers indicate statistical significance (α = 0.05). [4] Questions regarding motivation for antibiotic use and antibiotic source were only answered by those caregivers that reported using antibiotics for their animal(s). [5]Household member use of antibiotics was determined based on caregiver response to whether or not their child in the study had taken antibiotics in the past 3 months and whether or not a household member had taken antibiotics in the past 3 months, the latter of which was only asked to those who reported having a household member with an illness or infection in the past 3 months.
(PDF)

**S5 Table. Adjusted odds ratios for 3GCR-MDR and 3GCR-XDR *E. coli* carriage in dogs based on household characteristic risk factors.** [1]3GCR-MDR and 3GCR-XDR *E. coli* were determined from isolates resistant to ceftriaxone. [2]Odds ratio. [3]95% confidence interval. Bolded numbers indicate statistical significance (α = 0.05). [4]Livestock units = (0.01) (number of chickens) + (0.30) (number of pigs) + (0.80) (number of cattle) + (0.10) (number of sheep) + (0.10) (number of goats) + (0.02) (number of rabbits) + (0.01) (number of guinea pigs) + (0.03) (number of ducks) + (0.03) (number of quail).
(PDF)

**S6 Table. Adjusted odds ratios for 3GCR-MDR and 3GCR-XDR *E. coli* carriage in dogs based on household animal care and antibiotic knowledge, attitudes, and practices (KAP).** [1]3GCR-MDR and 3GCR-XDR *E. coli* were determined from isolates resistant to ceftriaxone. [2]Odds ratio. [3]95% confidence interval. Bolded numbers indicate statistical significance (α = 0.05). [4]Questions regarding motivation for antibiotic use and antibiotic source were only answered by those caregivers that reported using antibiotics for their animal(s). [5]Household member use of antibiotics was determined based on caregiver response to whether or not their child in the study had taken antibiotics in the past 3 months and whether or not a household member had taken antibiotics in the past 3 months, the latter of which was only asked to those who reported having a household member with an illness or infection in the past 3 months.
(PDF)

**S7 Table. Unadjusted odds ratios for CR, ESBL-producing, 3GCR-MDR, and 3GCR-XDR *E. coli* carriage in food animals based on household characteristic risk factors.** [1]3GCR-MDR and 3GCR-XDR *E. coli* were determined from isolates resistant to ceftriaxone. [2]Odds ratio. [3]95% confidence interval. Bolded numbers indicate statistical significance (α = 0.05). [4]Livestock units = (0.01) (number of chickens) + (0.30) (number of pigs) + (0.80) (number of cattle) + (0.10) (number of sheep) + (0.10) (number of goats) + (0.02) (number of rabbits) + (0.01) (number of guinea pigs) + (0.03) (number of ducks) + (0.03) (number of quail).
(PDF)

**S8 Table. Unadjusted odds ratios for CR, ESBL-producing, 3GCR-MDR, and 3GCR-XDR *E. coli* carriage in food animals based on household animal care and antibiotic knowledge, attitudes, and practices (KAP) risk factors.** [1]3GCR-MDR and 3GCR-XDR *E. coli* were determined from isolates resistant to ceftriaxone. [2]Odds ratio. A (-) indicates a positivity violation that prevented odds ratio (OR) calculation. [3]95% confidence interval. Bolded numbers indicate statistical significance (α = 0.05). [4]Questions regarding motivation for antibiotic use and antibiotic source were only answered by those caregivers that reported using antibiotics for their animal(s). [5]Household member use of antibiotics was determined based on caregiver response to whether or not their child in the study had taken antibiotics in the past 3 months and whether or not a household member had taken antibiotics in the past 3 months, the latter of which was only asked to those who reported having a household member with an illness or infection in the past 3 months.
(PDF)

**S9 Table. Adjusted odds ratios for 3GCR-MDR, and 3GCR-XDR *E. coli* carriage in food animals based on household characteristic risk factors.** [1]3GCR-MDR and 3GCR-XDR *E. coli* were determined from isolates resistant to ceftriaxone. [2]Odds ratio. [3]95% confidence interval. Bolded numbers indicate statistical significance (α = 0.05). [4]Livestock units = (0.01) (number of chickens) + (0.30) (number of pigs) + (0.80) (number of cattle) + (0.10) (number of sheep) + (0.10) (number of goats) + (0.02) (number of rabbits) + (0.01) (number of guinea pigs) + (0.03) (number of ducks) + (0.03) (number of quail).
(PDF)

**S10 Table. Adjusted odds ratios for 3GCR-MDR and 3GCR-XDR *E. coli* carriage in food animals based on household animal care and antibiotic knowledge, attitudes, and practices (KAP) risk factors.** [1]3GCR-MDR and 3GCR-XDR *E. coli* were determined from isolates resistant to ceftriaxone. [2]Odds ratio. A (-) indicates a positivity violation prevented odds ratio (OR) calculation. [3]95% confidence interval. Bolded numbers indicate statistical significance (α = 0.05). [4]Questions regarding motivation for antibiotic use and antibiotic source were only answered by those caregivers that reported using antibiotics for their animal(s). [5]Household member use of antibiotics was determined based on caregiver response to whether or not their child in the study had taken antibiotics in the past 3 months and whether or not a household member had taken antibiotics in the past 3 months, the latter of which was only asked to those who reported having a household member with an illness or infection in the past 3 months.
(PDF)

## Acknowledgments

We greatly appreciate the assistance of the past and present Prevention of community-acquired antimicrobial resistance (PRISA) fieldwork team, including Gabriela Heredia Arias, Paul Barahona Bonilla, Deysi Parrales Chicaiza, Anahi Flores Enriquez, Cristian Garzon, Joel Barahona Garzon, Valeria Garzon, Barbara Baque Pisco, Josué Fernando Barahona Garzón, and Rommel Guevara Santander. We also thank the communities involved in the study and our colleagues at the Microbiology Institute at the Universidad San Francisco de Quito for their support in conducting this research.

## Author Contributions

**Conceptualization:** Heather K. Amato, Carlos Saraiva-Garcia, Gabriel Trueba, Jay P. Graham.

**Data curation:** Siena L. Mitman, Heather K. Amato, Carlos Saraiva-Garcia, Liseth Salinas, Paúl Cárdenas, Jay P. Graham.

**Formal analysis:** Siena L. Mitman, Heather K. Amato.

**Funding acquisition:** Siena L. Mitman, Paúl Cárdenas, Gabriel Trueba, Jay P. Graham.

**Investigation:** Siena L. Mitman, Heather K. Amato, Carlos Saraiva-Garcia, Fernanda Loayza, Liseth Salinas, Kathleen Kurowski, Rachel Marusinec, Diana Paredes, Paúl Cárdenas, Gabriel Trueba, Jay P. Graham.

**Methodology:** Siena L. Mitman, Heather K. Amato, Carlos Saraiva-Garcia, Fernanda Loayza, Liseth Salinas, Kathleen Kurowski, Rachel Marusinec, Diana Paredes, Paúl Cárdenas, Gabriel Trueba, Jay P. Graham.

**Project administration:** Carlos Saraiva-Garcia, Fernanda Loayza, Paúl Cárdenas, Gabriel Trueba, Jay P. Graham.

**Resources:** Gabriel Trueba, Jay P. Graham.

**Software:** Siena L. Mitman, Heather K. Amato.

**Supervision:** Paúl Cárdenas, Gabriel Trueba, Jay P. Graham.

**Writing – original draft:** Siena L. Mitman, Heather K. Amato, Carlos Saraiva-Garcia.

**Writing – review & editing:** Siena L. Mitman, Heather K. Amato, Carlos Saraiva-Garcia, Fernanda Loayza, Liseth Salinas, Kathleen Kurowski, Rachel Marusinec, Diana Paredes, Paúl Cárdenas, Gabriel Trueba, Jay P. Graham.

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
