## [Decision Letter · Decision Letter 0]

28 Oct 2021

PGPH-D-21-00544

Risk factors for third-generation cephalosporin-resistant and extended spectrum b-lactamase-producing Escherichia coli carriage in domestic animals of semirural parishes east of Quito, Ecuador

Dear Dr. Graham,

Thank you for submitting your manuscript to PLOS Global Public Health. After careful consideration, we feel that it has merit but does not fully meet PLOS Global Public Health’s publication criteria as it currently stands. Therefore, we invite you to submit a revised version of the manuscript that addresses the points raised during the review process.

We look forward to receiving your revised manuscript.

Kind regards,

Reginald Quansah, Ph.D.

Academic Editor

Journal Requirements:

1. Please include additional information regarding the survey or questionnaire used in the study and ensure that you have provided sufficient details that others could replicate the analyses. For instance, if you developed a questionnaire as part of this study and it is not under a copyright more restrictive than CC-BY, please include a copy, in both the original language and English, as Supporting Information.

2. If you have no competing interests to declare, please state "The authors have declared that no competing interests exist".

3. Thank you for providing the following URL in your Data Availability Statement: https://doi.org/10.6078/D1ZM6F

Unfortunately, we are unable to access the URL as is. Please provide an updated, functional URL or DOI for the data underlying your study. We will update your Data Availability statement on your behalf to reflect the information you provide. 

4. Please amend your detailed Financial Disclosure statement. This is published with the article, therefore should be completed in full sentences and contain the exact wording you wish to be published.

i). State the initials, alongside each funding source, of each author to receive each grant.

ii). State what role the funders took in the study. If the funders had no role in your study, please state: “The funders had no role in study design, data collection and analysis, decision to publish, or preparation of the manuscript.”

Additional Editor Comments (if provided):

Reviewers' comments:

Reviewer's Responses to Questions

**Comments to the Author**

1. Does this manuscript meet PLOS Global Public Health’s publication criteria? Is the manuscript technically sound, and do the data support the conclusions? The manuscript must describe methodologically and ethically rigorous research with conclusions that are appropriately drawn based on the data presented.

Reviewer #1: Yes

Reviewer #2: Yes

2. Has the statistical analysis been performed appropriately and rigorously?

Reviewer #1: Yes

Reviewer #2: Yes

3. Have the authors made all data underlying the findings in their manuscript fully available (please refer to the Data Availability Statement at the start of the manuscript PDF file)?

Reviewer #1: Yes

Reviewer #2: Yes

4. Is the manuscript presented in an intelligible fashion and written in standard English?

Reviewer #1: Yes

Reviewer #2: Yes

5. Review Comments to the Author

Reviewer #1: Very well written paper on a timely and interesting topic. The study design was clearly explained, and the discussion was engaging.

Some specific comments.

L106: “small scale food animals” is confusing. Try “food animals from small-scale production facilities/small farms”

L129-130: please provide more information on the study site. How were these parishes selected? What is the definition of a “parish”? What percentage of the parishes around Quito do these 7 represent? Etc.

L138: Please provide details on or a citation for the validation process

L148: please explain whether samples were collected during each cycle, or whether one sample/pooled samples was collected per household across all three time units. If samples were collected from each household at more than one time point, how were the outcomes treated? Ever positive for the ESBL E. coli? What if a sample was positive on one cycle then negative at another cycle?

L167: check grammar of this sentence.

L207-212: I am not familiar with this unit. Please provide a brief explanation for why it is used, and why it would not be more useful to simply include a tally of the number of animals on the farm.

L216: I don’t know the size of these parishes and how close the households were to each other, but I can imagine the outcomes would be affected by clustering within a parish or within a certain geographic location of that parish. Did you assess /account for any clustering in your models? If not, please consider including this as a limitation. Was adjustment for multiple comparisons performed?

Table 1: Please consider indicating in the table whether the distribution of these variables were significantly different across the cycles.

L280: how was “Access to veterinary care” defined?

L338: check your spelling of “cefazolin”

Table 5: Do you have information on whether any human household members took antibiotics in the recent past?

Table 6: is there a reason the variable “feces management” was not included in this table? Presumably food animals would have access to feces that were left to disintegrate in the yard, for example.

L472: avoid contractions

L603: define this acronym

Reviewer #2: This study sought to unravel risk factors associated with domestic animal colonization with ceftriaxone-resistant (CR) and Extended-spectrum β-lactamase (ESBL)-producing Escherichia coli in semirural parishes east of Quito, Ecuador, where small-scale food animal production is common. Overall, the manuscript is well written and of good quality. The methodology is sound and the data support the conclusions. However, there are a number of issues that needed to be addressed and corrected:

1. Grammatical Errors

There are a number of grammatical errors, which need to be corrected. Kindly go through the entire manuscript and correct them. For example:

Page 3, line 64-66: “However, third-generation cephalosporin resistant (3CGR) bacteria, including those than can produce extended-spectrum β-lactamases (ESBLs), are becoming increasingly common (2–5).”

Page 6, Line 116-120: “While risk factors for childhood colonization with ESBL E. coli in the region have been studied, specific household characteristics and caregiver antimicrobial knowledge, attitudes, and practices (KAP) that contribute to domestic animal colonization with 3GCR and ESBL E. coli have yet to be explored (63).”

2. From your statement below (Page 5 & 6, line 108-112), Is there a link between limited research and risk factors for antibiotic resistant E. coli carriage in domestic animals? What is the justification for such statement? See below

Page 5 & 6, line 108-112: “The limited research that exists regarding risk factors for AR E. coli carriage in animals in small-scale production settings suggests the specific risk factors contributing to domestic animal colonization with ESBL and other AR E. coli are likely context-dependent, warranting closer attention in specific settings where small-scale food animal production occurs (26,29,35).”

3. Page 6, line 130-131: You stated that data was collected in three cycles. However, it is not clear what constitute a cycle. Clearly define what constitute a cycle.

4. Page 7, line137: What is the knowledge, attitudes, and practices (KAP) survey about? Is it about antibiotic usage? The statement should be completed.

5. Page 7, line138-139: You have stated that the KAP survey questionnaire used in this study has been previously validated. Kindly provide reference to that effect.

6. Page 10, 216: For consistency, use the terms “univariable and multivariable” instead of “univariate and multivariable”.

7. From the data you have presented in Table 1, cat is one of the common domestic animals you have sampled in all the three sample collection cycles. However, you collected only one faecal sample from a cat. What were the hindrances to collecting cat faecal samples and how is this likely to affect the interpretation of your results?

8. You stated several times in your results section that you adjusted for confounders. However, you fail to mention the particular confounders you adjusted for in the text or table.

9. Antimicrobial is not synonymous to antibiotics. Antimicrobials include antibiotics, antifungals, antivirals etc. Your statement above is therefore not correct. Modify the phrase “correctly answering no”.

10. You did not link Table 5 to the appropriate text. Kindly do that.

11. Dogs at households that obtained antimicrobials from a pet food store had increased odds of ESBL E. coli (OR 6.83, 95% CI 1.32-35.35). Conversely, a purchase of antimicrobials from a pet food store (OR 0.47, 95% CI 0.25-0.89) resulted in lower odds of food animal CR E. coli carriage. It is important to provide explanation for this disparity.

12. Page 27, line 432-434: What is the reason for comparing the prevalence of CR E. coli and ESBL E. coli carriage among domestic animals with human carriage at the first instance, when your study is about domestic animals carriage? Are there no similar studies among domestic animals you can compare to before comparing to human carriage?

6. PLOS authors have the option to publish the peer review history of their article (what does this mean?). If published, this will include your full peer review and any attached files.

**Do you want your identity to be public for this peer review?** For information about this choice, including consent withdrawal, please see our Privacy Policy.

Reviewer #1: **Yes: **Laurel Redding

Reviewer #2: No

---

## [Editor Report · Decision Letter 1]

19 Jan 2022

Risk factors for third-generation cephalosporin-resistant and extended-spectrum β-lactamase-producing Escherichia coli carriage in domestic animals of semirural parishes east of Quito, Ecuador

PGPH-D-21-00544R1

Dear Dr. Graham,

We're pleased to inform you that your manuscript has been judged scientifically suitable for publication and will be formally accepted for publication once it meets all outstanding technical requirements.

Within one week, you'll receive an e-mail detailing the required amendments. When these have been addressed, you'll receive a formal acceptance letter and your manuscript will be scheduled for publication.

An invoice for payment will follow shortly after the formal acceptance. To ensure an efficient process, please log into Editorial Manager at https://www.editorialmanager.com/pgph/ click the 'Update My Information' link at the top of the page, and double check that your user information is up-to-date. If you have any billing related questions, please contact our Author Billing department directly at authorbilling@plos.org.

Kind regards,

Reginald Quansah, Ph.D.

Academic Editor